# Benchmarking commonly used software suites and analysis workflows for DIA proteomics and phosphoproteomics

Ronghui Lou ®[1,2,3,7], Ye Cao[3,4,7], Shanshan Li[1,7], Xiaoyu Lang[1,2,6], Yunxia Li ®[4], Yaoyang Zhang ®[4,5] ✉ & Wenqing Shui ®[1,2] ✉

A plethora of software suites and multiple classes of spectral libraries have been developed to enhance the depth and robustness of data-independent acquisition (DIA) data processing. However, how the combination of a DIA software tool and a spectral library impacts the outcome of DIA proteomics and phosphoproteomics data analysis has been rarely investigated using benchmark data that mimics biological complexity. In this study, we create DIA benchmark data sets simulating the regulation of thousands of proteins in a complex background, which are collected on both an Orbitrap and a timsTOF instruments. We evaluate four commonly used software suites (DIA-NN, Spectronaut, MaxDIA and Skyline) combined with seven different spectral libraries in global proteome analysis. Moreover, we assess their performances in analyzing phosphopeptide standards and TNF-α-induced phosphoproteome regulation. Our study provides a practical guidance on how to construct a robust data analysis pipeline for different proteomics studies implementing the DIA technique.

Data-independent acquisition (DIA) mass spectrometry (MS) has emerged as a powerful technology for proteomics research as it promises both deep proteome coverage and consistent and accurate protein quantification for large-scale study designs[1–4]. As opposed to the traditional data-dependent acquisition (DDA) which selects the most abundant precursor ions for further analysis, the mass spectrometer in DIA experiments cycles through a pre-defined set of precursor isolation windows within which all the precursors are consistently fragmented. Thus, DIA proteomics establishes a complete and quantitative digital map for the proteome to be studied[5].

However, co-isolation and co-fragmentation of multiple precursors in the same selection window produces inherently complex tandem MS spectra and multiplexed chromatograms, which poses a significant challenge for DIA data processing[6,7]. A panel of software

suites such as OpenSWATH, Skyline, DIA-Umpire, and EncyclopeDIA has been developed to address computational challenges in DIA data analysis using a peptide-centric or spectrum-centric approach[8–11]. Up till now, the commercial package Spectronaut[2] has been the most widely employed in various DIA proteomics studies due to its versatile options and ready-to-use features for less experienced users[6,12–15]. More recently, several open-access tools with advanced infrastructure and unique strengths have been developed to offer more flexibility and less running costs than commercial software. For example, DIA-NN exploits deep neural networks and new quantification and signal correction strategies to improve proteome identification and quantification as well as enable high-throughput DIA analysis[7,16]. MaxDIA provides an end-to-end DIA data analysis workflow embedded into the MaxQuant environment with new features to achieve deep proteome coverages

[1]iHuman Institute, ShanghaiTech University, Shanghai 201210, China. [2]School of Life Science and Technology, ShanghaiTech University, Shanghai 201210, China. [3]University of Chinese Academy of Sciences, Beijing 100049, China. [4]Interdisciplinary Research Center on Biology and Chemistry, Shanghai Institute of Organic Chemistry, Chinese Academy of Sciences, Shanghai 201210, China. [5]Shanghai Key Laboratory of Aging Studies, 100 Haike Rd., Shanghai 201210, China. [6]Present address: University of Chinese Academy of Sciences, Beijing 100049, China. [7]These authors contributed equally: Ronghui Lou, Ye Cao, Shanshan Li. ✉e-mail: zyy@sioc.ac.cn; shuiwq@shanghaitech.edu.cn

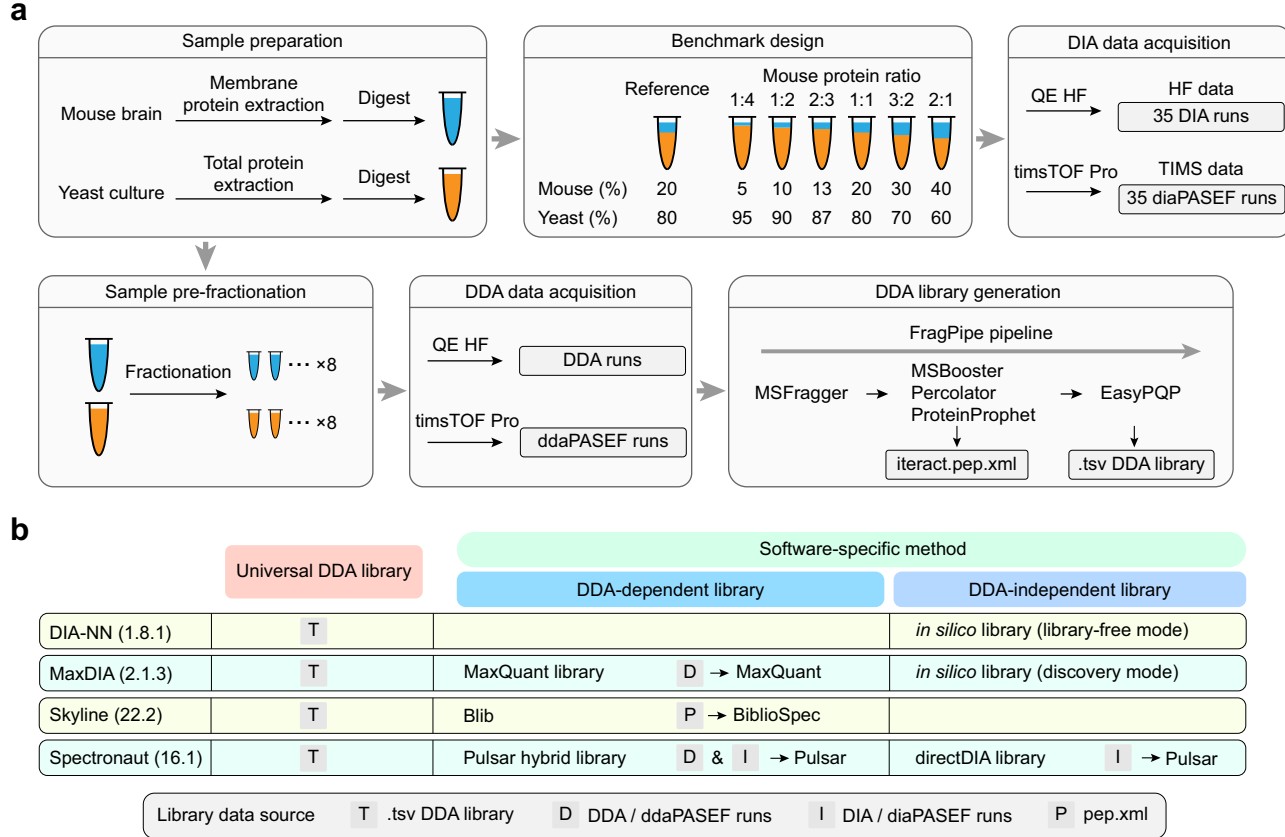

**Fig. 1 | Schematic of the benchmarking experiment and DIA data analysis workflows. a** Workflow of the benchmarking experiment. Mouse brain membrane protein digests were spiked into a yeast proteome background in seven defined proportions, yielding one reference and six mixtures with fixed mouse protein ratios relative to the reference. For each sample prepared in five replicates, DIA benchmark data were acquired on QE HF and timsTOF Pro instruments. Meanwhile DDA data were acquired on two instruments from pre-fractionated samples to build project-specific DDA libraries using the FragPipe pipeline. **b** Design of data analysis workflows evaluated in this study. The library generated by FragPipe serves as the universal library used by four software suites. Three software-specific DDA-dependent libraries were generated by MaxDIA, Skyline, and Spectronaut from different DDA and DIA data sources. Three DDA-independent libraries (two whole-proteome in silico libraries and one directDIA library) were generated and processed by DIA-NN, MaxDIA, and Spectronaut.

and consistent quantification under a reliable FDR control[17]. Despite various DIA software suites available, a consensus is lacking on which software is most suited to processing which type of proteomics data.

A DIA data analysis workflow will become more versatile if incorporates different classes of spectral libraries. Routinely, a project-specific DDA library is built from DDA data acquired on pre-fractionated samples or repeated injections. This experimental DDA library built from analysis of the same DDA data by different software tools such as MSFragger[18] and MaxQuant[19] could vary in their size and composition, which would substantially affect DIA data analysis results[20-22]. Alternatively, Spectronaut allows for the construction of a hybrid library which combines a project-specific DDA library with a directDIA library built from DIA data alone[15,23]. Recently, in silico libraries generated through predictions of fragment ion intensity and retention time for peptide sequences derived from the entire proteome or a targeted protein family using deep learning tools are gaining momentum[21,24-27].

A seminal work by Schilling et al. suggests that both the selection of software suites and the design of spectral libraries strongly impact the outcome of a DIA data analysis workflow[22]. However, the combination of DIA software and spectral libraries has been rarely investigated using benchmark data that mimics real biological complexity. Two recent studies have made significant progress in evaluating DIA analysis workflows using benchmark data specifically designed to reflect either the fluctuation of a small set of proteins or the background heterogeneity of clinical samples[22,28]. Of note, both studies

acquired data on Orbitrap-series instruments to evaluate different bioinformatics workflows for global proteome profiling.

In this study, we created DIA benchmark data sets simulating the regulation of thousands of proteins in a complex background, which were collected on both an Orbitrap instrument and a timsTOF instrument. With a unique feature of four-dimensional ion detection leading to superior speed and sensitivity, timsTOF represents a promising platform for broad proteomics application and merits specific investigation[29-31]. We evaluated four commonly used DIA software suites (DIA-NN[7], Spectronaut[2], MaxDIA[32], and Skyline[9]) in combination with different types of spectral libraries through the analysis of not only global proteomics data but also phosphoproteomics data. Our benchmark study reveals the distinct advantages of DIA-NN and Spectronaut in analyzing different types of DIA proteomics data acquired on state-of-the-art instruments.

## Results

### Design of the benchmarking experiment

To generate a benchmark sample set simulating systematic regulation of a large protein population, we prepared mouse brain membrane proteins spiked into a yeast proteome background in defined proportions (Fig. 1a). Relative to one hybrid proteome sample referred to as reference, the other six samples with different compositions yield expected mouse membrane protein ratios from 1:4 to 2:1. We designed this sample set with a relatively small magnitude of fold changes to assess the sensitivity of software suites in selecting differentially

expressed proteins (DEPs) above a commonly applied 1.5-fold threshold. Each benchmark sample was prepared in five process replicates and analyzed on two instrument platforms, QE HF in DIA mode and timsTOF Pro in diaPASEF mode (Fig. 1a). The resulting two benchmark data sets (HF data and TIMS data for short, 35 runs in each set) containing thousands of DEPs with defined ratios enabled comprehensive evaluation of multiple analysis workflows with different software suites. In the meantime, we performed DDA analysis of fractionated mouse membrane proteome and yeast proteome samples on QE HF and timsTOF Pro instruments. The DDA data was used to build project-specific libraries typically required for DIA data mining (Fig. 1a).

Because the size, quality, and composition of a spectral library have a profound impact on DIA data analysis, we built three classes of libraries to be tested with each software tool (Fig. 1b). A universal library was generated from the raw DDA data using a FragPipe pipeline[33]. Data analysis with the universal library by four software tools established an identical baseline for evaluation. Alternatively, Spectronaut, MaxDIA, and Skyline allow for processing the DDA data with an integrated search engine to generate software-specific DDA-dependent libraries. The universal library comprised 174,115 peptide precursors mapped to 11,725 proteins based on HF DDA data, and 225,350 precursors mapped to 13,704 proteins based on TIMS DDA data, which was modestly larger than or similar to different software-specific libraries (Supplementary Fig. 1a–h). In addition, an in silico library can be generated and exploited by DIA-NN in library-free mode or by MaxDIA in discovery mode, which circumvents the need for an experimental DDA library for DIA data analysis. The in silico library built from the mouse and yeast protein sequence databases comprised 1,529,467 peptide entries mapped to 23,812 proteins (Supplementary Fig. 1i). Spectronaut also supports library construction in a DDA-independent manner, through building a directDIA library in a much smaller size from DIA data alone (Supplementary Fig.1a, b, e, f). In total, we aim to exploit four software suites combined with seven spectral libraries (one universal library, three software-specific DDA-dependent libraries, and three DDA-independent libraries), resulting in 10 different data analysis workflows (Fig. 1b).

## Performance of proteome identification

We first implemented DIA-NN, Spectronaut, MaxDIA, and Skyline (all in the latest version) to process the HF data set. Although each software may assemble protein groups in a different way, we found the number of protein identifications re-assigned based on the razor protein inference[19] was very close to that reported by the software which was then used directly for comparison (Supplementary Fig. 2). With the universal library, DIA-NN, Skyline and Spectronaut yielded comparable coverages of the mouse membrane proteome (4919–5173 proteins) yet the FDR control by Skyline was insufficient (see the FDR assessment session for details). With a software-specific DDA-dependent library, Spectronaut attained the highest coverage by reporting 5354 mouse proteins and 67,310 peptides (Fig. 2a, Supplementary Fig. 3a). When utilizing an in silico library, DIA-NN achieved the best identification performance by reporting 5186 mouse proteins and 51,313 peptides (Fig. 2a, Supplementary Fig. 3a). Overlaps of mouse protein identifications by four software suites equipped with different libraries are summarized in Supplementary Figure 3. Notably, DIA-NN with the in silico library covered 94.3% proteins identified by itself with the universal library (Supplementary Fig. 3c). Similar numbers of peptides per protein were obtained by DIA-NN, Spectronaut, and MaxDIA, yet the density plot was skewed to the lower end for Skyline which reported the least peptide identifications (Fig. 2b).

Analysis of the TIMS benchmark data set with all four software suites gave rise to substantially expanded proteome coverages, due to the enhanced sensitivity of the instrument implementing a novel PASEF scan mode[29,30,34]. Both DIA-NN and Spectronaut gave a high performance by reporting 7128 and 7116 mouse proteins respectively

using the universal library (Fig. 2d, Supplementary Fig. 3d–f). Although DIA-NN combined with the in silico library marginally reduced the mouse proteome coverage compared to the universal library, this workflow still considerably exceeded the coverages attained by most other workflows (Fig. 2d, Supplementary Fig. 3d). We further analyzed the sub-proteome coverage of G protein-coupled receptors which are under-represented in most global proteomic surveys (e.g., 63 and 71 GPCR proteins reported from mouse and human brain tissues, respectively[35,36]). Remarkably, 127 and 123 GPCR identifications were yielded from TIMS data analysis by DIA-NN and Spectronaut, respectively, with the universal library, and 112 GPCR identifications by DIA-NN with the in silico library (Supplementary Fig. 4). Given that mouse proteins accounted for only 5–40% of total protein mass in the hybrid proteome samples with GPCRs even in a much smaller subpopulation, DIA-NN and Spectronaut both demonstrated superior capability to detect low-abundance proteins in a highly complex proteomic background.

The proteome coverage is commonly determined by the total number of identified proteins concatenated from all replicates yet not all proteins are identified in each replicate, thus leading to an issue of data incompleteness. DIA MS acquisition is renowned for enhanced data completeness by reducing missing values of protein intensities in the data matrix[1–3,16,37]. Not surprisingly, there is a negative correlation between the cumulative percentage of missing values for mouse proteins and the protein intensity rank for all analysis workflows (Fig. 2c, f). Among them, Spectronaut with a directDIA library yielded the highest data completeness (7.2% and 4.5% missing values across 35 runs for two data sets). MaxDIA with the universal library or DDA-dependent library also achieved high completeness for HF data (17.0% and 12.7% missing values) yet less completeness for TIMS data (21.4% and 20.2% missing values). DIA-NN with different types of libraries yielded similar and acceptable data completeness (16.6–18.7% missing values for two data sets). Data analysis by Skyline with the universal library or DDA-dependent library resulted in the lowest data completeness (36.9–40.7% missing values for two data sets) (Fig. 2c, f). Consistently, Skyline-based workflows reported the fewest mouse proteins with intensity measured in all five replicates compared to the other three tools (Supplementary Fig. 5).

## False positive and false negative rate assessment

A challenge in proteomic benchmark studies is that each software scores identifications and controls the false discovery rate (FDR) in different manners. Specifically, DIA-NN implements a fully connected neural network as the scoring model to discriminate targets and decoys[7]. Spectronaut develops Avalon based on a gradient boosting machine and exploits deep learning for scoring peptide-XIC matches[2,38]. MaxDIA employs XGBoost as its scoring function[32]. Skyline integrates mProphet to score peptide identifications using a linear model[9,39]. Additionally, DIA-NN and Spectronaut allow for FDR controls on both precursor and protein levels[2,7]. MaxDIA scores and filters library-to-sample matches prior to the protein-level FDR control[32], whereas Skyline controls FDR based on a mProphet-calculated detection $Z$ score[9]. For an objective assessment of FDR control, we employed a two-species library approach to estimate FDRs independent of the software. Leveraging a deep neutral network that was refined with our DDA data sets, we generated a predicted decoy library based on previously identified peptide precursors from the *Arabidopsis* proteome[40,41]. Appending incremental fractions of this decoy library (10%, 20%, 50%, and 100%) to the universal library to create a series of target-decoy libraries allowed us to assess not only FDRs but also false negative rates (FNRs) in proteomic identification (Supplementary Fig. 6a).

We calculated the percentage of *Arabidopsis* proteins or precursors out of all those identified with the target-decoy library as a proxy for FDR, which is termed the false ID percentage. Skyline, which

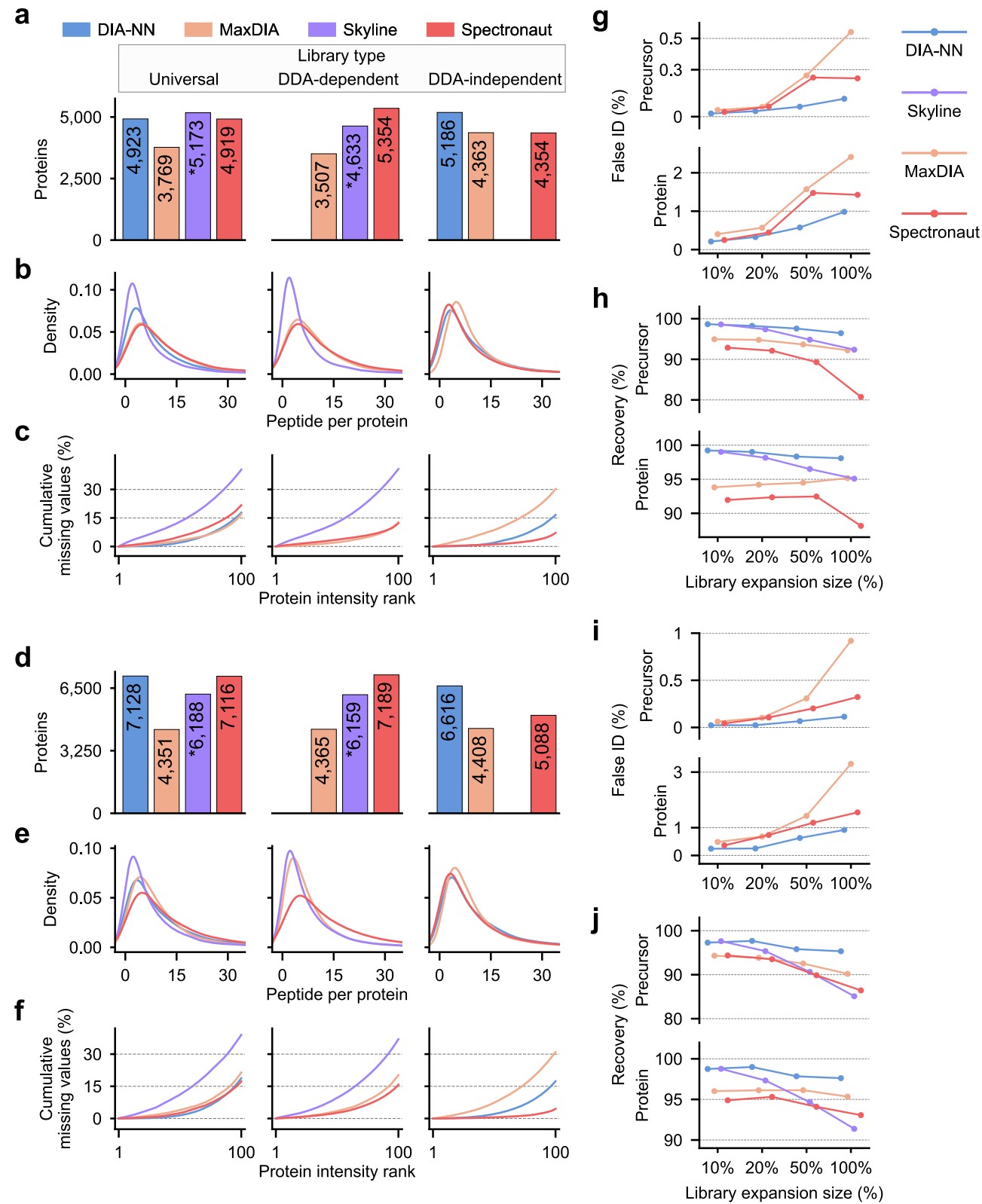

does not control a protein-level FDR, gave rise to abnormally high false ID percentages (6.9% and 9.5% for HF and TIMS data) when 10% decoy library was appended (Supplementary Fig. 6b). Estimated FDR control by MaxDIA was less stringent in processing TIMS data (3.3% and 0.92% false ID on protein and precursor levels) than HF data (2.4% and 0.54% false ID on protein and precursor levels) (Fig. 2g, i). DIA-NN and Spectronaut exerted equally adequate FDR control (<1.5% and 0.32%

false ID on protein and precursor levels) in processing both data sets, with DIA-NN modestly outperforming Spectronaut (Fig. 2g, i). When the search space was inflated by adding proportions of the decoy library, DIA-NN consistently recovered >98.1% mouse and yeast proteins and more than 96.4% peptide precursors initially identified with the universal library alone from HF data (Fig. 2h). It indicates that a low FNR is maintained by DIA-NN when a project-specific library contains

**Fig. 2 | Evaluation of proteome identification and FDR/FNR assessment.**
**a** Number of mouse protein identifications with different analysis workflows from HF data. The star symbol indicates the lack of protein-level FDR control by Skyline. **b** Kernel density estimate of the number of identified peptides per mouse protein from HF data. **c** Percentage of cumulative missing values as a function of mouse protein intensity rank (normalized to a scale of 1-100) for HF data. By ranking the protein intensity in a descending order, the number of missing values is cumulated and transformed into a percentage. **d–f** Same as **a–c** but for TIMS data. Results are summarized for each software equipped with a specific library as defined in Fig. 1b. **g** Percentage of false *Arabidopsis* precursor identifications (upper panel) and protein identifications (lower panel) from HF data analysis by different software with a series of target-decoy libraries. The percentage of false identifications (false ID) yielded by each software was used as a proxy for FDRs. **h** Percentage of recovered mouse and yeast precursors (upper panel) and proteins (lower panel) from HF data analysis by different software with a series of target-decoy libraries. Libraries are constructed by appending incremental fractions of the *Arabidopsis* decoy library (indicated in the *x* axis) to the universal library. The percentage of precursor and protein identifications yielded with a target-decoy library over that with the universal library alone is referred to as the recovery rate. The higher recovery rate indicates a lower FNR. **i, j** Same as **g**, **h** but for TIMS data. Source data are provided at https://doi.org/10.5281/zenodo.7409391.

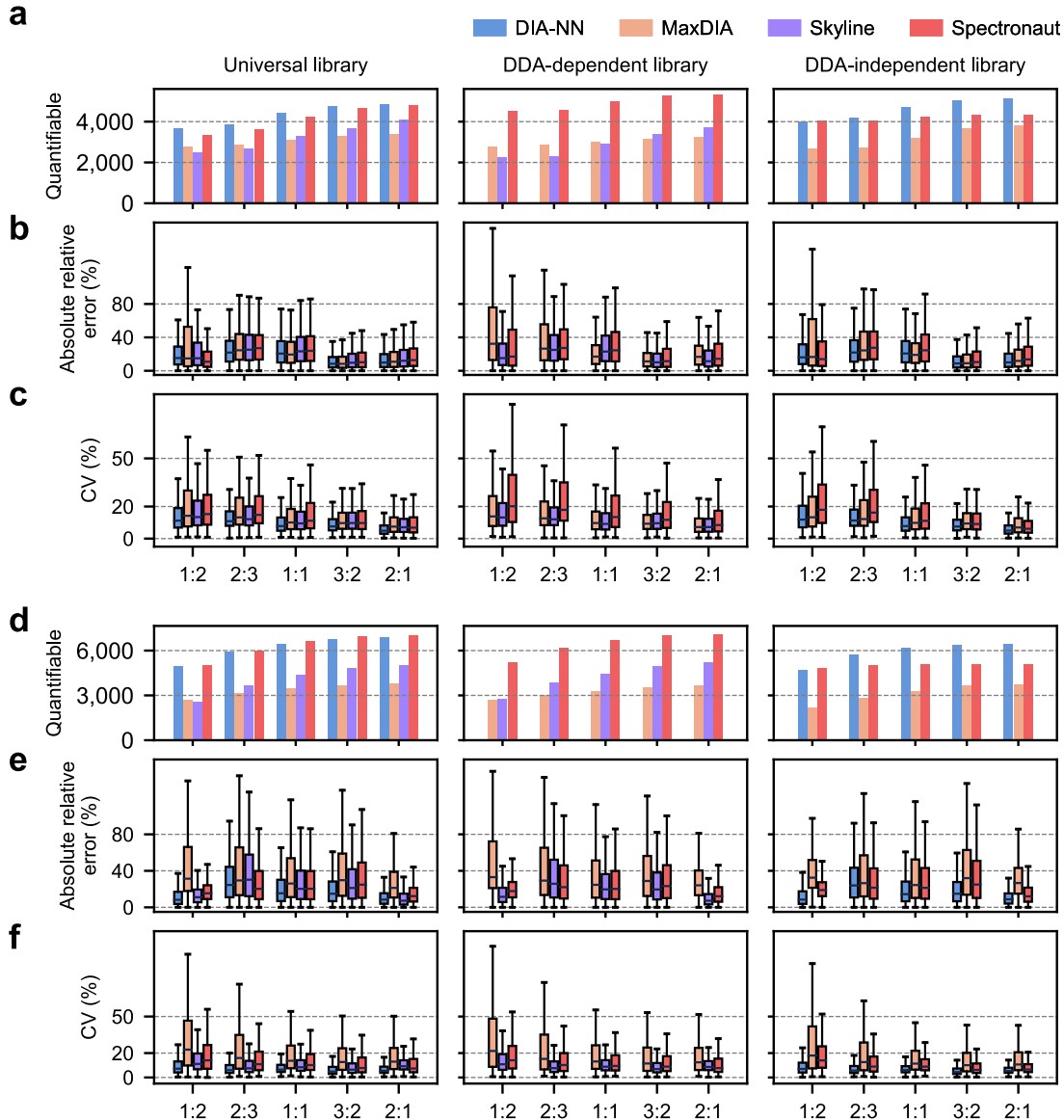

**Fig. 3 | Evaluation of quantification performance. a** Number of quantifiable mouse proteins (quantified in at least three of five replicates) yielded by different analysis workflows under each condition with the expected protein ratio from 1:2 to 2:1 for HF data. Results are summarized for each software equipped with a specific library as defined in Fig. 1b. **b** Distribution of absolute relative errors between expected and measured protein ratios by different analysis workflows under each condition for HF data. Boxplot center line, median; box limits, upper and lower quartiles; whiskers, 1.5× interquartile range. **c** Distribution of coefficient of variation (CV) for mouse proteins quantified by different analysis workflows under each condition for HF data. Boxplot center line, median; box limits, upper and lower quartiles; whiskers, 1.5× interquartile range. **d–f** Same as **a–c** but for TIMS data. Source data are provided at https://doi.org/10.5281/zenodo.7409391.

varying fractions of interference precursors that are not present in the sample. By contrast, data analysis using Spectronaut was less resistant to search space expansion, with 88.2% total proteins and 80.7% precursors retained when the full decoy library was appended (Fig. 2h).

Analysis of TIMS data by DIA-NN also showed strong stability by recovering more than 97.6% total proteins and 95.3% precursors during search space expansion, whereas analysis by Spectronaut reduced the proteome recovery rate, suggesting a higher FNR (Fig. 2j).

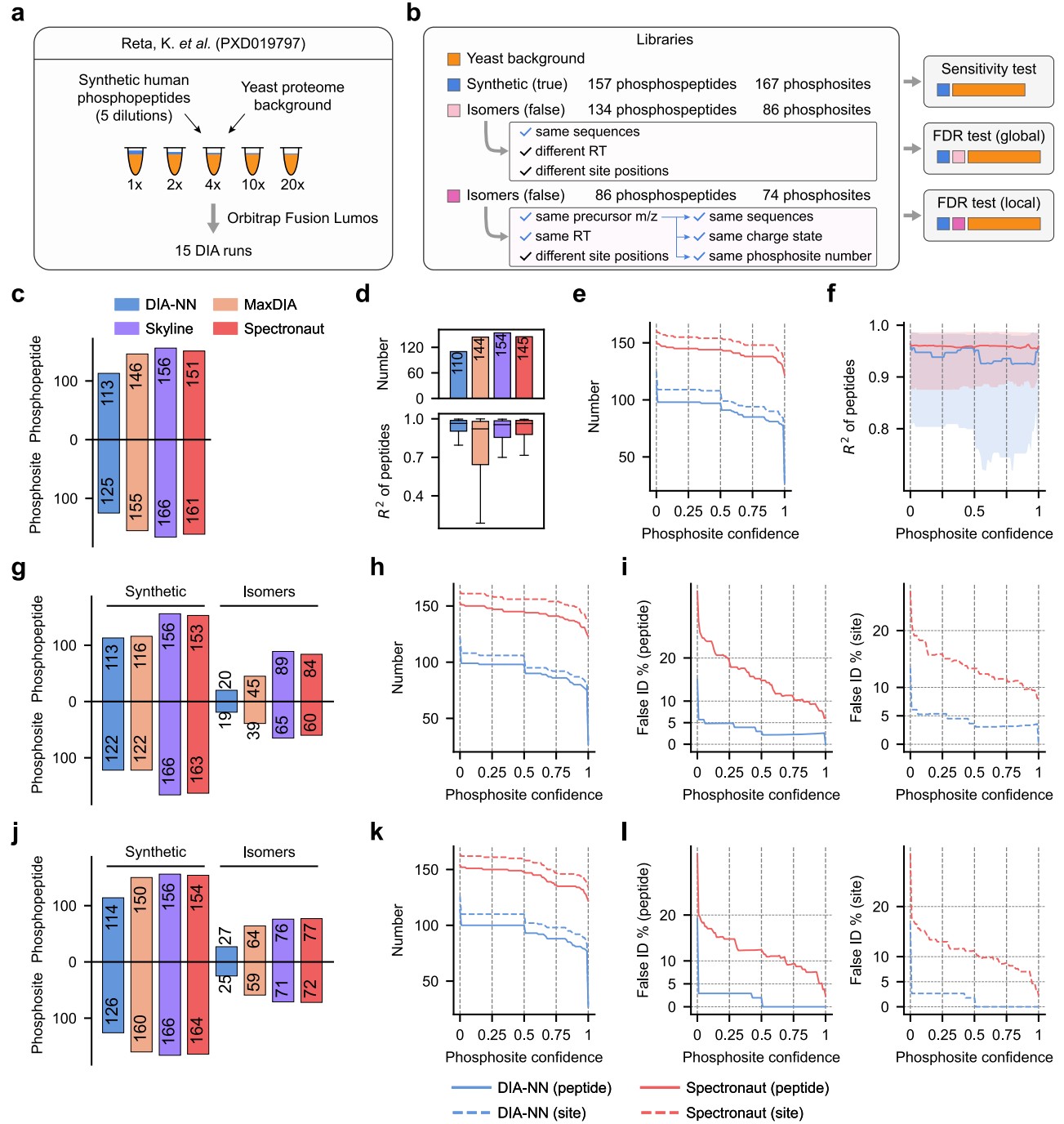

**Fig. 4 | Evaluation of phosphopeptide identification and phosphosite localization.** **a** A phosphopeptide benchmark data set was obtained from synthetic human phosphopeptides spiked into a yeast proteome background at five dilution concentrations. **b** Design of spectral libraries for three tests. See text and methods for details. **c** Number of identified synthetic phosphopeptides and phosphosites (true hits) in the sensitivity test. **d** Quantification linearity of identified synthetic phosphopeptides across the dilution series. The valid numbers of quantified proteins are shown in the upper panel. **e** Number of phosphopeptides and phosphosites as a function of the phosphosite confidence score cutoff. For peptides with multi-phosphosites, all sites need to pass the score cutoff. **f** Quantification linearity of phosphopeptides as a function of the phosphosite confidence score cutoff. The solid line indicates median values, with the interquartile range filled in light color. **g** Number of identified synthetic phosphopeptides and phosphosites (true hits) and isomers (false hits) in the global FDR test. **h** Same as **e** but for the global FDR test. **i** Estimated FDR on the peptide level (left) and site level (right) as a function of the phosphosite confidence score cutoff in the global FDR test. **j**–**l** Same as **g**–**i** but for the local FDR test. **e**, **f**, **h**, **i**, **k**, **l** The red and blue lines indicate results from Spectronaut and DIA-NN, respectively, on peptide (solid lines) and site (dashed lines) levels. Source data are provided at https://doi.org/10.5281/zenodo.7409391.

Given that DIA-NN analysis with the in silico library achieved a deeper or equivalent proteome coverage compared to the universal library, we performed FDR and FNR assessment by creating a target-decoy library that incorporated an in silico library generated from the *Arabidopsis* protein sequence database. Estimated FDRs and FNRs for the in silico library were as strictly restricted as for the universal library (<1.77% false ID, >98.65% proteome recovery) (Supplementary Fig. 6c). Taken together, DIA-NN enables the best control of both false positive and negative rates in processing benchmark data sets with either the universal library or the in silico

library, while three other software suites diminished their performance in at least one aspect of error rate assessment.

## Performance of proteome quantification and detection of differentially expressed proteins

To evaluate quantification performance based on original data, we analyzed search reports of precursor intensities generated by different analysis workflows without performing normalization, data imputation, or sparsity reduction. Previous studies demonstrated that data post-processing such as normalization and missing value imputation had minor or even negative impacts on quantification results when dealing with originally high-quality DIA data[22,28]. As for deriving protein intensities from precursor intensities, we tested three widely used methods (Top3, software built-in algorithm, and package *iq*[42] re-implementing algorithm MaxLFQ[43]) to select the best one for each software suite based on quantification precision and robustness (Supplementary Figs. 7, 8). Proteins with valid intensity values in at least three out of five replicates for each sample were considered quantifiable proteins and retained for further analysis. In line with the proteome identification coverage, DIA-NN and Spectronaut equipped with three classes of libraries all gave rise to more quantifiable mouse proteins than MaxDIA and Skyline (Fig. 3a, d).

The benchmark study design allowed us to precisely assess the quantification accuracy for mouse membrane proteins serially spiked into a complex yeast proteome background. Because the most diluted sample only comprised 5% mouse proteins and showed unusually high quantification variance (Supplementary Figs. 9a and 10a), we ignored this condition and compared results from the remaining five hybrid proteome samples with an expected mouse protein ratio from 1:2 to 2:1. In HF data analysis, DIA-NN achieved the best accuracy in ratio determination with three classes of libraries (median absolute relative errors of 8.1–21.9%) compared to the other suites (9.6–25.2% for Skyline, 10.2–27.5% for Spectronaut, and 8.3–27.5% for MaxDIA). A narrower distribution of relative errors was also observed for DIA-NN than the other suites at most conditions (Fig. 3b). Concordantly, the highest reproducibility of mouse protein quantification was achieved by DIA-NN with different libraries (median CVs of 4.9-11.8% and maximum interquartile range (IQR) of 13.5%), yet MaxDIA and Spectronaut suffered from a lower quantification reproducibility with a wider spread of CVs (median CVs of 6.9–14.2%, maximum IQR of 22.4% for MaxDIA; median CVs of 6.1–20.2%, maximum IQR of 29.4% for Spectronaut) (Fig. 3c). In TIMS data analysis, the best accuracy and reproducibility of protein quantification was also obtained by DIA-NN with different libraries (median absolute relative errors of 8.2–24.5%, median CVs of 4.4–7.2%). In contrast, protein ratios determined by MaxDIA and Spectronaut showed significantly larger variations among proteins or replicates at most conditions (Fig. 3e, f). Similar results were obtained in the assessment of peptide quantification by different workflows (Supplementary Figs. 9b, 10b). In addition, we compared the Pearson correlation of protein intensities between replicates. Protein quantification by DIA-NN showed the highest correlation (median correlation coefficients of 0.997-0.999) for all comparisons, confirming its strongest performance in quantification consistency (Supplementary Figs. 9c, 10c).

As the ultimate goal of the most proteomic analysis is to detect DEPs between two or multiple conditions, we assessed the sensitivity and specificity of different workflows in DEP detection using our benchmark data sets. In the pairwise comparison of any hybrid proteome sample with the reference, DEPs were extracted using the widely applied criteria of a fold change >1.5 and an adjusted *p*-value <0.05 (Limma reported). For the comparison with an expected protein ratio of 2:1, analysis using DIA-NN and Spectronaut with the universal library, in silico library or directDIA library achieved similarly high sensitivity of DEP detection from both HF data (87.5–90.2% quantified proteins as DEPs) and TIMS data (91.5–95.2% quantified proteins as

DEPs) (Supplementary Fig. 11). In TIMS data analysis, Skyline achieved an equally high sensitivity in DEP detection with different libraries (91.0–92.4% quantified proteins as DEPs) (Supplementary Fig. 11b). For another comparison with an expected protein ratio of 3:2, HF data analysis using Spectronaut resulted in a more sensitive detection of DEPs than the other suites whereas TIMS data analysis using all four suites demonstrated a comparable sensitivity of DEP detection (58.2–69.5% quantified proteins) (Supplementary Fig. 11). It implies more than half of proteins with an expected 1.5-fold change were selected when applying the exact fold-change threshold together with *p* value restriction. On the other side, in the pairwise comparison with an expected ratio of 1:1, HF data analysis using DIA-NN, Spectronaut, and Skyline resulted in detection of false DEPs at comparable rates (18.1–22.2%). However, in TIMS data analysis, DIA-NN yielded much smaller fractions of falsely detected DEPs (10.1–11.0%) than the other three suites (16.2–21.6%), indicating their lower specificity than DIA-NN (Supplementary Fig. 11).

Furthermore, we assessed the robustness of DEP detection using another approach based on the receiver operating characteristic (ROC) curve analysis[22,28], which led to the same conclusion (Supplementary Fig. 12). In summary, workflows based on DIA-NN enable DEP detection from our benchmark data sets with the best combination of sensitivity and specificity.

## Phosphopeptide identification, site localization, and stoichiometry measurement

To evaluate the performance of different workflows in phosphoproteomics data analysis, we first used a DIA data set acquired on synthetic human phosphopeptides spiked into a yeast proteome background at five different doses[14]. This data set comprised 157 detectable phosphopeptides containing 167 defined phosphosites (Fig. 4a). We performed three tests with different libraries to assess the sensitivity, error rates of phosphopeptide identification and site localization, and linearity of phosphopeptide quantification by different software suites.

In the first sensitivity test, we built a pure target library by merging the synthetic peptide library containing spectra of 157 phosphopeptides with a yeast tryptic peptide DDA library (Fig. 4b). Analysis of the DIA data with the pure target library by Skyline yielded the highest identification rate (156 phosphopeptides with 166 phosphosites detected) followed by Spectronaut while DIN-NN reported the lowest identification number without restricting any site localization confidence (Fig. 4c). Moreover, a similarly high linearity of phosphopeptide quantification over a dilution series was achieved by DIA-NN (median $R^2$ = 0.964) and Spectronaut (median $R^2$ = 0.963) (Fig. 4d). As both DIA-NN and Spectronaut report confidence scores for localized phosphosites, we plotted the number of phosphopeptide identification and linearity of quantification as a function of the score cutoff (Fig. 4e, f). Unlike Spectronaut which yielded smooth curves of the identification number and quantification linearity over a wide range of score cutoffs, DIA-NN sharply altered its performance in both aspects at several inflections (Fig. 4e, f).

Next, we performed global and local FDR tests with two decoy libraries consisting of false phosphopeptides to estimate FDRs of peptide identification and site localization by each tool. One decoy library comprised 134 isomeric phosphopeptides containing 86 phosphosites which shared the same sequences as the synthetic phosphopeptides yet with different phosphosite positions. The sequences, modification sites, and charge states of decoy phosphopeptides were retrieved from a public human phosphoproteome database[44], and their MSMS spectra and iRT were predicted by the DeepPhospho model[21]. Data analysis using this decoy library appended to the pure target library allowed us to estimate FDRs for phosphopeptide and phosphosite identification on a global level. Although Skyline yielded the most phosphopeptide identifications, it reported

the highest number of false isomers, leading to an FDR of 36.3% and 28.1% on peptide and site levels, suggesting the least reliability of its intrinsic error rate control without site confidence restriction (Fig. 4g). Intrinsic site-level FDRs were 13.5% for DIA-NN, 22.3% for MaxDIA and 26.9% for Spectronaut, respectively (Fig. 4g). Importantly, by adjusting the site confidence score cutoff, global FDRs on phosphopeptide and site levels dropped to 5.7% and 6.1% respectively for DIA-NN with a score cutoff as low as 0.01, and 10.8% and 11.5% for Spectronaut with a score cutoff of 0.75 (Fig. 4i).

Another decoy library comprised 86 isomeric phosphopeptides containing 74 phosphosites which shared the same sequences, charge states, and the exact numbers of phosphosites as the synthetic phosphopeptides yet only differed in the site position (Fig. 4b). Data analysis using this decoy library appended to the pure target library allowed us to estimate the local FDRs for phosphopeptide and phosphosite identification. Although this type of positional isomers represents the most challenging decoys to be distinguished from true targets, the sensitivity of detecting synthetic phosphopeptides by DIA-NN and Spectronaut was not compromised as the peptide identification curves displayed the same patterns as those obtained with the previous two libraries (Fig. 4e, h, k). Surprisingly, local site-level FDR of data analysis with DIA-NN abruptly dropped to 2.7% and even to 0% when the site score confidence was set to 0.01 and 0.51, respectively. By contrast, a smoother FDR curve was observed for Spectronaut with the site-level local FDR reaching 8.7% and 3.5% under a score cutoff of 0.75 and 0.99, respectively (Fig. 4l). We also performed the sensitivity and global FDR tests on Spectronaut and DIA-NN using DDA-independent libraries, and observed a very similar trend (Supplementary Fig. 13). In summary, compared to Spectronaut which affords higher sensitivity in phosphopeptide detection with a less acceptable FDR, DIA-NN exerts a more stringent FDR control at an expense of the authentic phosphopeptide identification rate.

Analysis of phosphorylation site stoichiometry can provide unique insights into cell signaling regulation and facilitate the identification of functional phosphosites[45,46]. Accurate stoichiometry measurement depends on both the accuracy and precision of phosphoproteomic quantification. To further benchmark the quantification performance, we analyzed a hybrid proteome data set with fixed phosphopeptide stoichiometries from 1% to 99%[46]. After extracting the quantification data of phosphopeptides, non-phosphopeptides, and corresponding proteins, we implemented a 3D multiple regression model-based approach for site-specific stoichiometry calculation[46]. Stoichiometry measurement based on the quantification results by DIA-NN and Spectronaut using a project-specific DDA library yielded equally high accuracy and similar precision for all stoichiometry levels and across a wide range of phosphosite score cut-offs (Supplementary Fig. 14).

### DIA phosphoproteomics data analysis in a biological setting

To evaluate DIA-NN and Spectronaut in a cellular signaling study, we performed a TNF-α-induced phosphoproteomics experiment in which MCF-7 cells were stimulated with TNF-α in the absence or presence of an I-kappa-B kinase inhibitor TPCA-1 (Fig. 5a). Then we acquired DIA and DDA data for each replicate on both QE HF-X and timsTOF Pro instruments. The DIA data sets were processed by two software tools using either a project-specific DDA library or DDA-independent libraries (Fig. 5a).

For each software, we tested a regular and a stringent phosphosite localization score cut-offs as inferred from the synthetic phosphopeptide analysis (0.01 and 0.51 for DIA-NN, 0.75 and 0.99 for Spectronaut). Contrary to the synthetic phosphopeptide analysis, DIA-NN reported more phosphosites than Spectronaut for most comparisons under the regular or stringent score cutoff, except for the in silico library in HF-X data analysis (Supplementary Fig. 15a, b). Analogous to the proteomics benchmark data analysis, smaller variations of

phosphosite quantification were observed for HF data analysis by DIA-NN than Spectronaut. The difference in quantification consistency between two software became unappreciable for TIMS data analysis (Supplementary Fig. 15c, d).

Given the context of TNF-α-induced signaling, we then selected up-regulated phosphosites (>1.5-fold change and Limma reported $p < 0.05$) to assess how much biological insight can be gained into the signaling network. In HF-X data analysis, DIA-NN with the project-specific DDA library under the regular cutoff yielded the largest number of up-regulated sites (183 sites) whereas in TIMS data analysis, Spectronaut with the directDIA library under the regular cutoff yielded the highest number (254 sites) (Fig. 5b). In accordance with the previous study[47], the combined treatment with TNF-a and TPCA-1 resulted in a predominant inhibition of TNF-α-dependent phosphorylation (>84% for HF-X data and >90% for TIMS data) by different analysis workflows (Fig. 5b). More importantly, signaling pathway analysis based on TNF-α-induced sites revealed the capability of each analysis workflow to recapitulate the known TNF-α signaling network. In HF-X data analysis, three well-characterized pathways in response to TNF-α stimulation were most significantly enriched by DIA-NN with the in silico library under the regular cutoff. However, in TIMS data analysis, Spectronaut gave the best performance in the enrichment of five known pathways when equipped with the directDIA library under the regular cutoff (Fig. 5c). Consistently, DIA-NN analysis of HF-X data reported more known TNF-α-induced phosphosites within the TNF-α signaling pathway while Spectronaut analysis of TIMS data revealed more known phosphosites (Fig. 5d).

In summary, the bioinformatics analysis based on TNF-α-regulated sites indicated the preference of DIA-NN in processing HF-X phosphoproteomics data and Spectronaut in processing TIMS data. Furthermore, the use of a DDA-independent library by both software exhibited comparable or even slightly better performance than the project-specific DDA library in the identification of known TNF-α-dependent phosphosites and signaling pathways, probably due to the relatively small size of the DDA library generated in this study (Supplementary Fig. 15e). As the above results were obtained based on phosphosite quantification data with missing value imputation, we assessed the completeness of detected phosphosites and recovery of the signaling network from the original data without imputation (Supplementary Figs. 16, 17). More known TNF-α signaling pathways in line with a higher number of TNF-α-responding phosphosites can be recovered from the imputed data, suggesting imputation using an appropriate algorithm could facilitate DIA phosphoproteomics analysis.

## Discussion

Compared to the two recent elegant studies that have provided a comprehensive comparison of DIA software tools and workflows[22,28], our study design has several distinct features. First, our benchmark samples contain a large number of DEPs (e.g., 5168 and 6553 mouse proteins quantified in at least three replicates by DIA-NN with the in silico library from HF and TIMS data sets), allowing for more in-depth evaluation of quantification performance. Second, we additionally investigated the performance and robustness of DIA bioinformatic workflows in phosphoproteomics data analysis. Third, our result would provide a rich resource for the use and optimization of software tools specific for DIA proteomics on a timsTOF instrument which is gaining wider popularity[31,48–51]. Fourth, the software panel evaluated in our study includes MaxDIA, which was most recently developed and regarded as a landmark platform for DIA data mining[32]. Fifth, our results were obtained using all software in their latest versions (Fig. 1b), which are evidently different from earlier versions used in published studies, especially for Spectronaut (Supplementary Note 1).

Collectively, our study reveals which combination of software and spectral library is preferred for analyzing global proteomics or

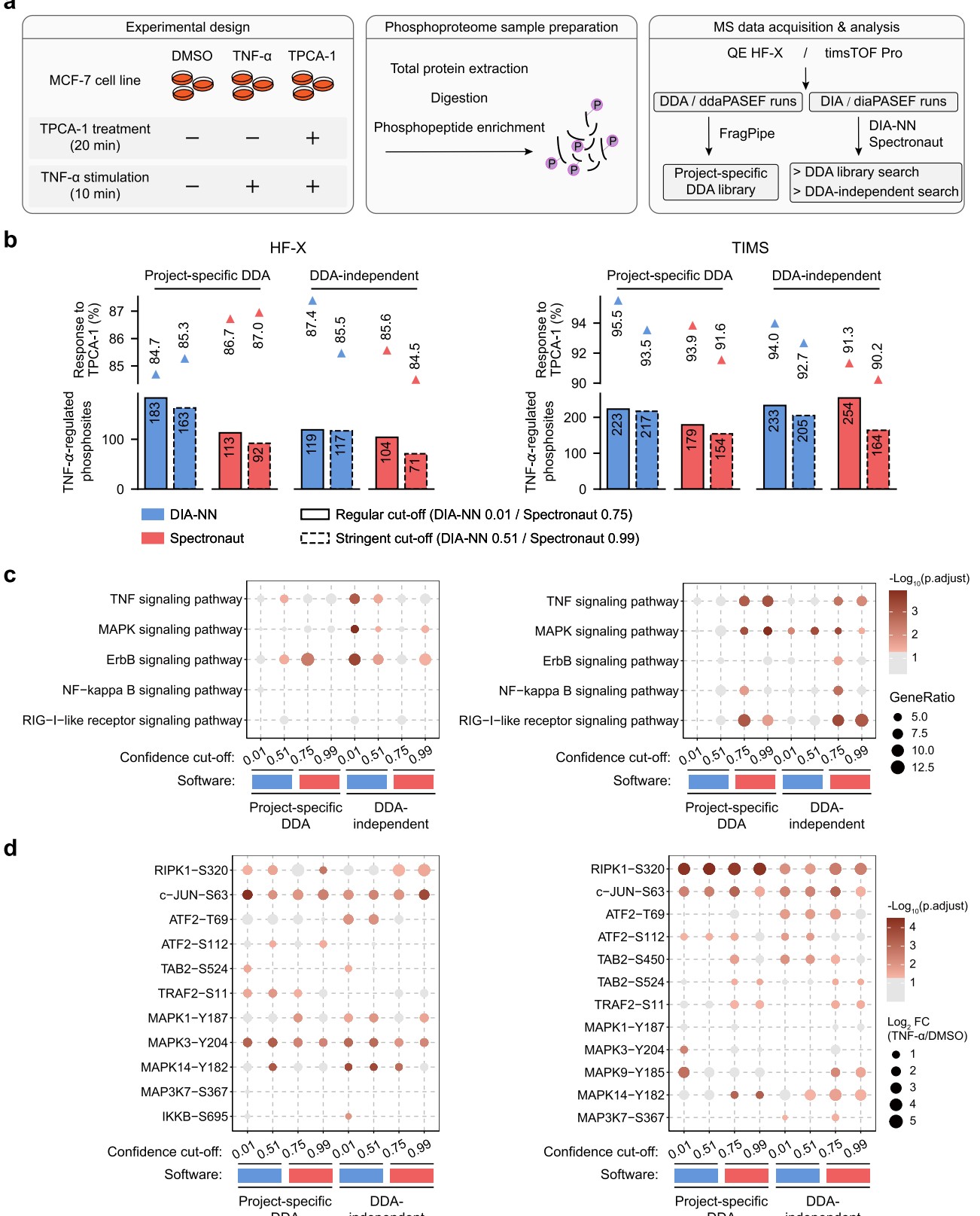

phosphoproteomics data. In our proteomics benchmark data analysis, while Spectronaut reported slightly more protein identifications with the universal or DDA-dependent libraries, DIA-NN yielded a higher proteome coverage with a DDA-independent library (Fig. 2). Moreover, DIA-NN gave better performance than Spectronaut in the FDR/FNR control, quantification accuracy and precision, as well as sensitivity

and specificity of DEP detection for most comparisons (Figs. 2, and 3, Supplementary Figs. 11, 12). Given the overall superior performance and the open-access feature, we would recommend DIA-NN for global DIA proteomics data analysis. Of note, previous studies have demonstrated that in silico libraries built on proteome-scale protein sequence databases face a key challenge of extensive query space, which would

**Fig. 5 | Comparison of DIA-NN and Spectronaut in TNF-α-induced phospho-proteome data analysis. a** Experimental scheme. MCF-7 cells were stimulated with TNF-α in the absence or presence of an I-kappa-B kinase inhibitor TPCA-1 (each condition in biological triplicate). For each replicate, DDA and DIA data were acquired on QE HF-X and timsTOF Pro instruments. The DIA data sets were processed by two software tools using either a project-specific DDA library or a DDA-independent library. **b** Number of TNF-α-regulated phosphosites from the analysis of HF-X data (left) and TIMS data (right) with different workflows. Phosphosites in response to TPCA-1 are those up-regulated by TNF-α and suppressed by TPCA-1 treatment, and their percentages over all up-regulated sites by TNF-α are indicated. **c** Enriched KEGG pathways based on the analysis of HF-X data (left) and TIMS data (right) with different workflows. Significantly enriched pathways (adjusted $p < 0.05$) are annotated in a color gradient. **d** Phophosites upregulated by TNF-α and included in the TNF-α pathway. They were identified in the analysis of HF-X data (left) and TIMS data (right) with different workflows. Significantly regulated sites (FC > 1.5, adjusted $p < 0.05$) are annotated in a color gradient. Results in **b**–**d** are based on phosphosite quantification data with missing value imputation. Source data are provided at https://doi.org/10.5281/zenodo.7409391.

cause reduced detection sensitivity and increased false positives for DIA data analysis with Spectronaut[21,52]. However, our study proved the ability of DIA-NN and MaxDIA to attain a comparable or even higher proteome coverage with an adequate FDR control when using a whole-proteome in silico library *versus* an extensive experimental DDA library. Thus, an in silico library-based DIA analysis workflow is recommended for global proteomic studies of various organisms, which gives excellent performance in a most economic fashion.

The superior performance of DIA-NN for in-depth proteomic profiling could be attributed to its improved peak selection algorithm and a scoring model exploiting a fully connected neural network that involves 73 subscores[7]. Spectronaut v16 substantially improves the sensitivity and specificity of protein identification by introducing a new machine learning framework Avalon based on gradient boosting. Notably, analysis of HF and TIMS data with Spectronaut and MaxDIA using different classes of libraries gave rise to higher protein/peptide quantification variation than DIA-NN and Skyline, which compromised their sensitivity in true DEP detection. Nevertheless, MaxDIA allows for data analysis with diverse experimental and in silico libraries, and provides convenience to users familiar with the MaxQuant platform. Although Skyline lacks an FDR control at the protein level, it excels in peptide peak extraction and quantification from DIA data. We performed extensive parameter optimization and tests for each software with our benchmark data to highlight critical settings (Supplementary Notes 1–4).

Apart from the global DIA proteomics, our study conducted a systematic evaluation of DIA software suites and workflows for phosphoproteomics data analysis, which unveiled the complementary performance of DIA-NN and Spectronaut. In the synthetic phospho-peptide data analysis, while Spectronaut showed higher sensitivity in phosphopeptide and phosphosite detection, DIA-NN excelled in the FDR control on both peptide and site levels. Importantly, unlike most previous studies defining class-I sites based on a non-discriminant confidence score cutoff of 0.75[53,54], we proposed regular and stringent phosphosite confidence thresholds that are optimized for specific software (0.01/0.51 for DIA-NN, 0.75/0.99 for Spectronaut) to balance the identification rate and the error rate. Interestingly, a much smaller difference in the proteome coverage was observed between DIA-NN and Spectronaut in the analysis of a real-case phosphosignaling data set. Comparison of two software in the TNF-α-induced phosphoproteomics analysis further revealed their differential behaviors in the enrichment of signaling pathways and discovery of regulated phosphosites from HF-X data *versus* TIMS data. Therefore, users can select a combination of software, library and score cutoff to analyze their own DIA phosphoproteomics data, depending on the data type, the library completeness, and their preference of coverage or error rate control.

In conclusion, we envision this study would provide practical guidance on the analysis of different types of DIA proteomics data acquired on state-of-the-art instruments. Nevertheless, we recognized the four software suites are undergoing intensive and continuous development, and their respective stronger or weaker performances may change in the future. Thus, we hope our study would also offer software developers useful data and information to benchmark new algorithms or improve existing ones. Bioinformatics advances are expected to constantly drive DIA-based proteomics research.

## Methods

### Mouse brain membrane protein preparation and digestion

The C57BL/6 mice (Shanghai Jiesijie, Laboratory Animal Technology Company, China) were housed under a 12-hour light-dark cycle with free access to water and food. All experimental mice were male adults (9–10 weeks of age) and habituated for 1 week at least before the experiments. The mice were euthanized with 2% chloral hydrate and rapidly dissected to obtain the brain tissue. All experimental procedures were approved by the Institutional Animal Care and Use Committee at ShanghaiTech University and performed in accordance with National Institutes of Health guidelines. Membrane fraction was isolated from mouse brain tissue according to our previous work[12]. Briefly, brain tissue was homogenized in the isolation buffer of 30 mM Tris-HCl (pH 7.4), 0.1 mM EDTA, 0.5% BSA, 300 mM sucrose with a protease inhibitor cocktail (Roche). The homogenate was centrifuged for 15 min at $3000 \times g$ at 4 °C. Cell pellet was collected and resuspended in isolation buffer and homogenized again and centrifuged at $10,000 \times g$ at 4 °C for 20 min. Supernatant was ultra-centrifuged at $160,000 \times g$ at 4 °C for 1 h. The membrane pellet was washed with 100 mM $Na_2CO_3$, 1 M KCl, and 100 mM Tris-HCl (pH 7.4) separately and ultra-centrifuged for 1 h at $160,000 \times g$ at 4 °C. The membrane pellet was resuspended in lysis buffer of 5% SDC, 50 mM $NH_4HCO_3$ and protein concentration was determined by BCA assay (TIANGEN, Beijing, China).

Mouse membrane proteins were reduced with 15 mM dithio-threitol (DTT) for 30 min at 56 °C, and alkylated with 40 mM iodoacetamide for 30 min at room temperature in darkness, additional 25 mM DTT was added to consume the excess iodoacetamide. Protein samples were diluted with 50 mM $NH_4HCO_3$ to 1% SDC. Sequencing-grade trypsin (Promega, Madison, USA) was added at an enzyme-to-protein ratio of 1:50 (w/w) and incubated for 3 h at 37 °C, and additional trypsin at a 1:100 (w/w) ratio was added for digestion overnight at 37 °C. Digestion was quenched with 1% FA. Peptides were desalted with Oasis HLB cartridge (Waters) and lyophilized under vacuum and stored at −80 °C. Five replicates were prepared, each starting with 20 μg mouse membrane protein extracts.

### Yeast total cell protein preparation and digestion

*Saccharomyces cerevisiae* BY4742 strain was grown at 30 °C to mid-log-phase in YPD medium (OXOID, UK). Cells were harvested by centrifugation at $4000 \times g$ for 5 min and washed twice with ice cold PBS. Cell pellets were resuspended in a lysis buffer of 5% SDC and 50 mM $NH_4HCO_3$ with a protease inhibitor cocktail (Roche). After adding an equal amount of glass beads, the cells were lysed by glass bead beating and lysates were centrifuged at $13,000 \times g$ for 10 min. The supernatants were collected, and protein concentration was determined using BCA assay (TIANGEN, Beijing, China). Yeast protein digestion and desalting were performed in the same way as mouse protein samples. Five replicates were prepared, each starting with 60 μg yeast protein lysates.

### Peptide mixing and pre-fractionation

After re-dissolving in 0.1% FA, each replicate of mouse membrane protein digest was spiked into one replicate of yeast protein digest to generate one reference (containing 20% mouse membrane proteome)

and six samples for comparison (containing 5%, 10%, 13%, 20%, 30% and 40% mouse membrane proteome). Pairwise comparison of each sample with the reference yields theoretical mouse protein ratios of 1:4, 1:2, 2:3, 1:1, 3:2, and 2:1. All hybrid proteome samples were spiked in with an iRT reference kit (Biognosys, Zurich, Switzerland) and prepared in five replicates.

For experimental DDA library generation, protein digests from mouse membrane fractions or yeast cells were separately loaded onto a high-pH RP fractionation spin column (Pierce) according to the manual instruction. Peptides bound to the column were washed once with water. A step gradient of acetonitrile (5–50%) in a volatile high-pH elution solution was applied to elute peptides into eight fractions sequentially. All fractions were dried under vacuum and stored at −80 °C.

### Sample preparation for TNF-α-induced phosphoproteomics

MCF-7 cells (Cobioer Biosciences Co., Ltd., Nanjing, China) were treated with DMSO, or 10 ng/ml of TNF-α for 10 min, or 10 μM of TPCA-1 for 20 min followed by TNF-α for 10 min. After washing cells with PBS three times, cells were lysed in RIPA buffer containing 1% Triton X-100 (v/v), 0.5% (w/v) SDS, 50 mM Tris-HCl (pH 7.4), 150 mM sodium chloride, 2 mM sodium orthovanadate, 5 mM sodium fluoride and protease inhibitors cocktail (Roche). Lysates were sonicated for 2 min and centrifuged at $13,000 \times g$ for 10 min at 4 °C. Proteins were precipitated by adding 4× volumes of cold acetone. The resulting protein pellets were washed with acetone, dissolved in 8 M urea, 100 mM Tris-HCl (pH 8.5), and subjected to a standard in-solution digestion with trypsin (Promega). The digests were acidified with formic acid and desalted using a C18 column.

Phosphorylated peptides were enriched using $TiO_2$ resin (GL Sciences Inc.). 2 mg of peptides for each condition was dissolved in 8 mL loading buffer containing 1 M glycolic acid, 5% TFA, and 80% acetonitrile. 6 mg of $TiO_2$ beads suspended in 800 μL acetonitrile was loaded to C8 tips. Peptide solutions were loaded to $TiO_2$ tips and washed with 200 μL loading buffer for five times. Tips were washed by 200 μL washing buffer I containing 5% TFA and 80% acetonitrile, and washing buffer II containing 1% TFA and 80% acetonitrile for three times, respectively. Tips were subsequently washed with 200 μL Milli-Q water twice. The bound phosphorylated peptides were eluted with 200 μL elution buffer I containing 1.8% ammonia hydroxide in water twice, and 200 μL elution buffer II containing 6% ammonia hydroxide in acetonitrile. Elution was collected for vacuum centrifuge to dryness. Each sample was prepared in three biological replicates.

### MS data acquisition on QE HF and HF-X

**DDA data acquisition.** The fractionated peptide samples were dissolved in 0.1% FA and analyzed using an EASY-nLC 1200 system (Thermo Fisher Scientific) coupled to a Q-Exactive HF mass spectrometer (Thermo Fisher Scientific). The fractions were separated on an analytical column (200 mm × 75 μm) in-house packed with C18-AQ 1.9 μm C18 resin (Dr. Maisch GmbH, Germany) with a gradient of 3–8% solvent B (0.1% FA/80% ACN) in 25 min, 8–20% B in 70 min, 20–42% B in 25 min, 42–100% B in 4 min and 100% B for 6 min at a flow rate of 300 nl/min. MS acquisition parameters were set as follows: the full scan range, 300–1650 m/z; MS1 resolution, 60,000; AGC, 3e6; maximum injection time, 20 ms; top 15 precursors selected for subsequent MS2 scans; MS2 resolution, 15,000; AGC, 1e5; maximum injection time, 25 ms; isolation window, 1.4 m/z; normalized collision energy (NCE), 27%; dynamic exclusion time, 30 s.

Phosphopeptide samples were analyzed using an EASY-nLC 1200 system (Thermo Fisher Scientific) coupled to a Q-Exactive HF-X mass spectrometer (Thermo Fisher Scientific). The fractions were separated on an analytical column (180 mm × 100 μm) in-house packed with C18-AQ 1.9 μm C18 resin (Dr. Maisch GmbH, Germany) with a gradient of 5–10% solvent B (0.1% FA/80% ACN) in 4 min, 10–27%

B in 88 min, 27–37% B in 21 min, 37–100% B in 4 min and 100% B for 3 min at a flow rate of 300 nL/min. MS acquisition parameters were set as follows: the full scan range, 350–1500 m/z; MS1 resolution, 60,000; AGC, 3e6; maximum injection time, 20 ms; top 20 precursors selected for subsequent MS2 scans; MS2 resolution, 30,000; AGC, 1e5; maximum injection time, 45 ms; isolation window, 1.6 m/z; NCE, 27%; dynamic exclusion time, 30 s.

**DIA data acquisition.** The mouse-yeast hybrid proteome samples were separated using an EASY-nLC 1200 system with the same gradient as described above. MS data acquisition in DIA mode was performed on Q-Exactive HF using 22 variable windows covering a mass range of 300–1300 m/z. The resolution was set to 120,000 for MS1 and 30,000 for MS2. The AGC was 3e6 in both MS1 and MS2, with a maximum injection time of 60 ms in MS1 and auto in MS2. NCE was set to 28%.

Phosphopeptide samples were separated using an EASY-nLC 1200 system with the same gradient as described above. MS data acquisition in DIA mode was performed on Q-Exactive HF-X using 30 variable windows covering a mass range of 350–1500 m/z. The resolution was set to 60,000 for MS1 and 15,000 for MS2. The AGC was 3e6 in MS1 and 5e5 in MS2, with a maximum injection time of 100 ms in MS1 and auto in MS2. NCE was set to 28%.

### MS data acquisition on timsTOF Pro

**DDA data acquisition.** The same fractionated peptide samples were analyzed using nanoElute LC system coupled to a timsTOF Pro mass spectrometer (Bruker, Bremen, Germany). The peptides were separated on an analytical column (250 mm × 75 μm 1.6 μm C18 resin, IonOpticks) with a gradient of 2–22% solvent B (0.1% FA in ACN) in 90 min, 22–37% B in 10 min, 37–80% B in 10 min and 80% B for 10 min at a flow rate of 300 nl/min. The dual TIMS analyzer was operated at a fixed duty cycle with a ramp time of 100 ms, and the total cycle time was 1.16 s. DDA was performed in PASEF mode with 10 PASEF scans per topN acquisition cycle in a mass range from 100 m/z to 1700 m/z with charge states from 0 (unassigned) to 5+. The ion mobility was scanned from 0.6 to 1.6 $Vs/cm^2$. Precursors that reached a target intensity of 20,000 were selected for fragmentation and dynamically excluded for 0.4 min (mass width 0.015 m/z, $1/K_0$ width 0.015 $Vs/cm^2$). The quadrupole isolation width was set to 2 m/z for m/z < 700 and to 3 m/z for m/z > 700, and the collision energy was linearly interpolated between $1/K_0$ values, from 20 eV at 1.6 $Vs/cm^2$ to 59 eV at 1.6 $Vs/cm^2$, keeping constant above or below. The TIMS elution voltage was calibrated linearly to obtain reduced ion mobility coefficients ($1/K_0$) using three selected ions of the Agilent ESI-L Tuning Mix (m/z 622, 922, 1222).

Phosphopeptide samples were analyzed using nanoElute LC system coupled to a timsTOF Pro mass spectrometer (Bruker, Bremen, Germany). The peptides were separated on an analytical column (250 mm × 75 μm 1.6 μm C18 resin, IonOpticks) with a gradient of 2–22% solvent B (0.1% FA in ACN) in 90 min, 22–37% B in 10 min, 37–80% B in 10 min and 80% B for 10 min at a flow rate of 300 nL/min. The dual TIMS analyzer was operated at a fixed duty cycle with a ramp time of 100 ms, and the total cycle time was 1.16 s. DDA was performed in PASEF mode with 10 PASEF scans per topN acquisition cycle in a mass range from 100 m/z to 1700 m/z with charge states from 0 (unassigned) to 5+. The ion mobility was scanned from 0.6 to 1.5 $Vs/cm^2$. Precursors that reached a target intensity of 5000 were selected for fragmentation and dynamically excluded for 0.4 min (mass width 0.015 m/z, $1/K_0$ width 0.015 $Vs/cm^2$). The quadrupole isolation width was set to 2 m/z for m/z < 700 and to 3 m/z for m/z > 700, and the collision energy was linearly interpolated between $1/K_0$ values, from 20 eV at 0.6 $Vs/cm^2$ to 59 eV at 1.6 $Vs/cm^2$, keeping constant above or below. The TIMS elution voltage was calibrated linearly to obtain reduced ion mobility coefficients ($1/K_0$) using three selected ions of the Agilent ESI-L Tuning Mix (m/z 622, 922, 1222).

 

**DIA data acquisition.** The mouse-yeast hybrid proteome samples were separated using a nanoElute LC system with the same gradient as described above. The MS data were acquired using the diaPASEF method. The capillary voltage was set to 1400 V. The MS and MS/MS spectra were acquired from 100 to 1700 m/z. The ion mobility was scanned from 0.6 to 1.6 Vs/cm². The ramp time was set to 100 ms. The collision energy was ramped linearly as a function of the mobility from 59 eV at $1/K_0 = 1.6$ Vs/cm² to 20 eV at $1/K_0 = 0.6$ Vs/cm². Collision energy was set to 10 eV to prevent fragmentation to visualize the isolation of precursor ions and analyze the ion current from multiply charged precursors. Isolation windows of a 25 m/z width were set to cover the mass range of 400 to 1200 m/z in diaPASEF.

Phosphopeptide samples were separated using a nanoElute LC system with the same gradient as described above. The MS data were acquired using the diaPASEF method. The capillary voltage was set to 1400 V. The MS and MS/MS spectra were acquired from 100 to 1700 m/z. The ion mobility was scanned from 0.75 to 1.3 Vs/cm². Precursors that reached a target intensity of 10,000 were selected for fragmentation and dynamically excluded for 0.5 min. The ramp time was set to 100 ms. The collision energy was ramped linearly as a function of the mobility from 59 eV at $1/K_0 = 1.6$ Vs/cm² to 20 eV at $1/K_0 = 0.6$ Vs/cm². Isolation windows of a 25 m/z width were set to cover the mass range of 452 to 1177 m/z in diaPASEF.

### Synthetic phosphopeptide benchmark data set

The synthetic phosphopeptide data set was downloaded from JPOST with identifier JPST000859 (ProteomeXchange code PXD019797). It was acquired from five mixtures of synthetic human phosphopeptides spiked into tryptic peptides from yeast whole-cell lysates[14]. Specifically, 166 human phosphopeptides were synthesized and mixed with the yeast proteome background at dilution ratios of 1×, 2×, 4×, 10×, and 20×. These mixed samples were analyzed in injection triplicate on Orbitrap Fusion Lumos instrument in DIA mode. The spectral library for all synthetic phosphopeptides was also downloaded from JPOST. Of the 166 synthesized phosphopeptides, 157 peptides matching MS1 acquisition windows in the DIA experiment were retained in the library, which contains 167 unique phosphosites.

### Protein FASTAs

Mouse (organism ID 10090, 17082 reviewed entries), yeast (organism ID 559292, 6730 entries), and human (organism ID 9606, 20386 reviewed entries) protein sequences were all downloaded from UniProtKB version 2021_03[55]. *Arabidopsis* protein sequences were downloaded from UniProtKB version 2022_01 (organism ID 3702, 16202 reviewed entries)[55]. Species-specific protein FASTAs were used in all DDA/DIA data analysis.

### Library generation for DIA proteomics benchmark data analysis

**Universal spectral library.** DDA data acquired on QE HF and ddaPASEF data acquired on timsTOF Pro were used to generate two universal libraries by FragPipe[33] (version 17.1) using a pre-defined workflow DIA_SpecLib_Quant. Specifically, decoys were first added to the FASTA which contains both mouse and yeast protein sequences. Then, MSFragger[18,56,57] (version 3.4) was used to search DDA raw data, with the following settings: precursor and fragment mass tolerance 20 ppm; strict trypsin with no more than two missed cleavages; peptide length 7–52; peptide mass 500–5000; C + 57.021464 as fixed modification; M + 15.9949 and N-term +42.0106 as variable modifications; min matched fragments 4; max fragment charge 2. In the validation step, MSBooster was implemented on both spectra and RT levels, and then Percolator[58] and ProteinProphet[59] integrated in Philosopher[60] (version 4.1.0) were used for PSM validation and protein inference. Library generation was conducted using EasyPQP (version 0.1.26) with RT calibration based on iRTs and Lowess fraction set to 0. Only fragment types b and y were included with a tolerance of 15 ppm.

**MaxQuant library for MaxDIA.** DDA or ddaPASEF data search results with MaxQuant[19] (version 2.1.3.0) were used to build the MaxQuant library which comprised three files (msms.txt, evidence.txt, and peptides.txt). MaxQuant parameters were set as follows: search type, Standard for DDA data or TIMS DDA for ddaPASEF data; TIMS half-width, TIMS step, and TIMS resolution set as default for ddaPASEF data; Carbamidomethyl on C as fixed modification; Oxidation on M and Acetyl at protein N-terminus as variable modifications; Trypsin/P and number of max missed cleavages 2; peptide mass tolerance in first search, 20 ppm; peptide mass tolerance in the main search, 4.5 ppm for HF data and 10 ppm for TIMS data; PSM and protein FDRs were both set to 0.01.

**Blib library for Skyline by BiblioSpec.** BiblioSpec processes an intermediate result file pep.xml from a common DDA data processing pipeline to generate the Blib library to be used by Skyline[9]. Here we built the Blib library based on the iteract.pep.xml files generated in the FragPipe pipeline. A detailed procedure is available on FragPipe GitHub page (https://github.com/Nesvilab/FragPipe/blob/gh-pages/docs/tutorial_skyline.md). In brief, the following files were extracted from the FragPipe working folder: 16 raw DDA or ddaPASEF data files; 16 uncalibrated mgf files; 16 interact.pep.xml files; the protein.fas file generated by Philosopher. Then, *Import DDA Peptide Search* wizard was used on the Skyline start page. After the iteract.pep.xml files were uploaded, a cutoff score was specified according to the FragPipe log file. iRT standard peptides were set to Biognosys-11 and all compatible variable modifications were selected. For Full-Scan settings, precursor charges were 2–6 for HF DDA data and 1–5 for TIMS DDA data, which was consistent with the charge state distribution in the universal library. Mass accuracy was 10 ppm and 15 ppm for HF DDA data and TIMS DDA data, respectively, and ion mobility resolving power was kept as default 30 for TIMS data. When importing FASTA, we used the optimized protein.fas from Philosopher, and set Trypsin [KR|P] as specific enzyme with no more than 2 missed cleavages.

**DirectDIA library and hybrid library for Spectronaut by Pulsar.** Pulsar embedded in Spectronaut[2] (version 16.1) was used to generate a directDIA library from DIA or diaPASEF data and a hybrid library by further merging the directDIA library and DDA library built from DDA or ddaPASEF data. In brief, both the directDIA library and DDA library were generated with the same settings and the hybrid library was generated by using the search archives of these two libraries. The settings for Pulsar and library generation were as followings: Trypsin/P as specific enzyme; peptide length from 7 to 52; max missed cleavages 2; toggle N-terminal M turned on; Carbamidomethyl on C as fixed modification; Oxidation on M and Acetyl at protein N-terminus as variable modifications; FDRs at PSM, peptide and protein level all set to 0.01; minimum fragment relative intensity 1%; 3–6 fragments kept for each precursor.

**In silico library for MaxDIA in discovery mode.** We generated in silico libraries using DIAtools[32] (commit 57f3977 on 25 Mar 2021, https://github.com/cox-labs/DIAtools) for data analysis by MaxDIA. MS spectral prediction was performed using Prosit[24] (commit dd16c47, https://github.com/kusterlab/prosit) on a local machine, with model weights downloaded from https://figshare.com/projects/Prosit/35582 (identifiers *Prosit - Model – Fragmentation* and *Prosit - Model – iRT*). At a pre-processing step, mouse and yeast protein sequences were digested to yield an input file for Prosit with default settings: CE 28; peptide length 7–30; precursor charge 2–3; missed cleavages 0–1. As CE varies in different acquisition windows for diaPASEF data, the CE value for each precursor was re-assigned based on precursor m/z. The Prosit input file was submitted by curl and output was set to a generic format. At a post-processing step, to prepare an RT file required by DIAtools for RT prediction, we downloaded the public discovery libraries pre-released

by MaxDIA for mouse and yeast proteomes from https://datashare.
biochem.mpg.de/s/qe1IqcKbz2j2Ruf?path=%2FDiscoveryLibraries to
be used in our DIA benchmark data analysis. The search results pro-
vided the RT input for DIAtools and specific hyper-parameters of the
RT model were set as follows: percentage of training data 70%; random
seed 0; batch size 512; learning rate 0.0005; learning rate decay for
Adam 1e-8; epochs 50. The output files from the post-processing script
were msms.txt, evidence.txt, and peptides.txt.

**Target-decoy library for FDR/FNR assessment**
**Arabidopsis peptide sequences.** To collect *Arabidopsis* peptide
sequences that are MS detectable, we downloaded *Arabidopsis* peptide
precursors from ProteomicsDB[40] (identifier PRDB004266 for project
Mergner_Nature_2020[41]). All identified peptides from each tissue in
this data set covering 12 total experiments were retrieved. These
peptide sequences were filtered based on the following criteria:
sequences included in the used version of *Arabidopsis* FASTA, peptide
length 7–52; maximum missed cleavages 2 by Trypsin/P; both excised
or un-excised protein N-terminal M were retained; peptides of the
same sequences as forward or reversed sequences from in silico
digestion of mouse or yeast proteome were removed.

**Target-decoy library generation.** We first generated two predicted
*Arabidopsis* spectral libraries as decoy libraries for the FDR/FNR
assessment using two benchmark data sets. Predicted libraries were
built on *Arabidopsis* peptide precursors using DeepPhospho[21] (commit
a779fd9 on 5 March 2022, https://github.com/weizhenFrank/
DeepPhospho), which contained predicted MSMS spectra and iRT
for HF data analysis and additional predicted ion mobility (1/K0) for
TIMS data analysis. In brief, the ion intensity and iRT models in
DeepPhospho were fine-tuned based on pre-trained model weights
(https://download.iprox.cn/IPX0003513000/IPX0003513001/
DeepPhosphoModels-PretrainParams.zip) with the universal library as
the training data. Main parameters for DeepPhospho runner were set
as follows: data split ratio 70:18:12; epoch 20; batch size for ion
intensity model and RT model, 128 and 256, respectively; initial
learning rate 1e-4; retention time scale from −90 to 160; ensemble
for RT model was used. We also built an ion mobility model by
performing transfer learning on the fine-tuned ion intensity model
for TIMS data. In brief, all modules in the ion intensity model
were loaded except the final FC layers which would output the tensor
with a shape of *peplen* × *ion_type*, and a new FC layer to aggregate
the output from the transformer module to a single value was initi-
alized. A demo script for ion mobility model training was released
in the commit a779fd9 of DeepPhospho. We trained the model
by fixing parameters in all modules except the newly initialized
FC layer (10 epochs) followed by fine-tuning all parameters (another
10 epochs).

After the two predicted *Arabidopsis* spectral libraries were
obtained, target-decoy libraries with expanding search spaces were
generated by merging the universal library and a defined proportion of
the *Arabidopsis* library (10%, 20%, 50%, and 100%). Sub-sampling of the
entire predicted *Arabidopsis* library was performed on the precursor
level, and precursors with m/z out of the MS1 acquisition windows of
HF or TIMS data sets were excluded.

**Library generation for synthetic phosphopeptide benchmark
data analysis**
**Library for the sensitivity test.** The synthetic human phosphopeptide
library, the yeast tryptic peptide library, and raw DIA data were
downloaded from JPOST with identifier JPST000859[14]. These two
libraries were then merged by Spectronaut to create a pure target
library, and exported in a plain text format. In the synthetic human
phosphopeptide library, 80 peptide precursors were removed as their
m/z exceeded the MS2 isolation windows of DIA data, which resulted in

a total of 157 synthetic phosphopeptides (222 precursors) in the final
target library.

**Library containing isomeric phosphopeptides for global and local
FDR tests.** To build decoy libraries for FDR tests, we extracted two sets
of isomeric phosphopeptide sequences from a published human
phosphoproteome database[44] according to different criteria. For the
global FDR test, phosphopeptide isomers need to share the same
peptide sequences as the synthetic human phosphopeptides but differ
in phosphosite number and/or localization, with phosphosite Ascore
>13 and phosphorylation as the only variable modification. The most
frequently detected charge state in the database was assigned to the
phosphopeptide isomer precursor. As a result, 134 isomeric phos-
phopeptide precursors containing 86 phosphosites were selected as
decoy hits for the global FDR test.

As for the local FDR test, phosphopeptide isomers need to share
the same peptide sequences and total phosphosite numbers as
the synthetic human phosphopeptides yet only differ in phosphosite
localization. In this case, each pair of a synthetic phosphopeptide
precursor and its positional isomer precursor should have identical m/
z, charge state, and RT, with only difference existing in their MSMS
spectra. A total of 86 isomeric phosphopeptide precursors containing
74 phosphosites were obtained as decoy hits for the local FDR test.

We fine-tuned DeepPhospho models using the yeast proteome
library (20 epochs with an initial learning rate 1e-4 and batch size 128
for both ion intensity and RT models) followed by the synthetic
phosphopeptide library (15 epochs with an initial learning rate 5e-5 and
batch size 16 for both models). Other hyper-parameters were set as
defaults in config files (commit a779fd9). Then we generated predicted
libraries based on the two sets of isomeric phosphopeptide precursors
described above using DeepPhospho models. 4–20 fragments with
minimum 5% relative intensity were retained for each precursor in the
library. The predicted libraries were appended to the pure target
library and used in two different FDR tests.

**Library generation for TNF-α-induced phosphoproteome data
analysis**
**Project-specific DDA library.** DDA data acquired on QE HF-X and
ddaPASEF data acquired on timsTOF Pro were used to generate two
project-specific libraries by FragPipe[33] (version 18.0, combined with
MSFragger 3.4, Philosopher 4.4.0, and EasyPQP 0.1.30) with the same
settings as the two benchmark data sets, except for these parameters:
STY + 79.96633 as an additional variable modification; max 5 variable
modifications on peptide; min matched fragments 5; specific losses
allowed in EasyPQP.

**DirectDIA library for Spectronaut by Pulsar.** Pulsar embedded in
Spectronaut[2] (version 16.1) was used to generate a directDIA library
from DIA or diaPASEF data with the same settings as the two bench-
mark data sets, except for these parameters: Phospho on STY as
additional variable modification; 4–12 fragments for each precursor.

**DIA data processing**
The raw HF and HF-X data (.raw files) and TIMS data (.d folders) were
directly imported into four software tools (DIA-NN, Spectronaut,
MaxDIA, and Skyline) without any format transformation.

**DIA-NN.** In the analysis of HF and TIMS data, search parameters of
DIA-NN[7] (version 1.8.1) were set as follows: precursor FDR 1%;
mass accuracy at MS1 and MS2 set to 5 ppm and 15 ppm for HF
data and both 0 for TIMS data; scan window set to 0; iso-
topologues and MBR turned on; protein inference at gene level;
heuristic protein inference enabled; quantification strategy set to
Robust LC (high precision); neural network classifier double-pass
mode; cross-run normalization off. The universal library was used

and protein re-annotation was performed. In the library-free mode, the main search settings were the same with additional settings for in silico library generation as follows: Trypsin/P with maximum 1 missed cleavage; protein N-terminal M excision on; Carbamidomethyl on C as fixed modification; no variable modification; peptide length from 7 to 30; precursor charge 1–4; precursor m/z from 300 to 1300 for HF data and 400 to 1200 for TIMS data; fragment m/z from 300 to 1800 for HF data 300 to 1700 for TIMS data. The search results were further filtered with $q$ value <0.01 for protein groups at the library level. In the analysis of synthetic phosphopeptide and TNF-α-induced phosphoproteome data, most search settings were the same except for these changes: neural network classifier set to single-pass mode; mass accuracy at MS1 and MS2 set to both 0 for synthetic phosphopeptide data, and 7.2 ppm and 25 ppm for HF-X and 15 ppm and 13.5 ppm for TIMS for TNF-α-induced phosphoproteome data; phosphorylation monitored; max 2 variable modifications on peptide in the library-free mode.

**MaxDIA.** In the analysis of HF and TIMS data, search parameters of MaxDIA[32] (version 2.1.3.0) were set as follows: Oxidation on M and Acetyl at protein N-terminus as variable modifications; Carbamidomethyl on C as fixed modification; digestion by Trypsin/P, with maximum two missed cleavages; protein quantification based on both unique and razor peptides; decoy generated by revert; FDR at any level kept at 0.01; DIA quantification method Mixed, LFQ split; ML for DIA on a global level; LFQ turned on and min ratio count of 2 without normalization; split protein groups by taxonomy ID turned on at species level. In the synthetic phosphopeptide data analysis, the search settings were the same except Carbamidomethyl on C and Phospho on STY set as variable modifications.

We further tested many parameter combinations in DIA benchmark data analysis, and found most of them had minor or no influence except transfer $q$ value which is the most sensitive parameter that can affect proteomic identification and quantification. In addition to the default value of 0.3 in MaxDIA, we tested four transfer $q$ values (0.05, 0.1, 0.15, 0.2) in HF data analysis with the MaxQuant library (Supplementary Note 4.1). Considering the proteome coverage and quantification robustness, we kept transfer $q$ of 0.3 when comparing MaxDIA with other software tools.

**Spectronaut.** In the analysis of HF and TIMS data, search parameters of Spectronaut[2] (version 16.1.220730.53000) were set as follows: mutation with NN predicted fragments to generate decoy; machine learning performed per run; precursor PEP cutoff 0.2; precursor $q$ value cutoff 0.01; protein $q$ value cutoff 0.01 at experiment level and 0.05 at run level; data filtering set to $Q$ value; cross-run normalization off. In the synthetic phosphopeptide and TNF-α-induced phosphoproteome data analysis, the search settings were the same except for single hit definition and minor grouping on modified sequences but not stripped sequences, cross-run normalization on, PTM localization on, and PTM probability cutoff set to 0.

**Skyline.** In the analysis of HF and TIMS data, search parameters of Skyline[9] (version 22.2.0.255) were set as follows: Trypsin [KR|P] as enzyme and maximum missed cleavages 2; peptide length 7–52; Carbamidomethyl on C as fixed modification; Acetyl at protein N-terminus, Oxidation on M, Ammonia Loss on K, N, Q, and R, and Water Loss on D, E, S, and T as variable modifications; precursor charge 2–6 for HF data and 1–5 for TIMS data to matches those in libraries; ion charge 1, 2; ion types y, b; product ion selection from ion 3 to last ion −1; ion match tolerance 0.05 m/z; 3–6 fragments picked from library; DIA as MS2 acquisition method, centroided with 8.6 ppm mass accuracy for HF data and 15 ppm

for TIMS data; isolation window directly extracted by importing the raw data; retention time filtering within 12 min and 10 min predicted RT for HF data and TIMS data, respectively. Search parameters for synthetic phosphopeptide data analysis are as follows: Carbamidomethyl on C and Phospho on ST and Y as variable modifications; precursor charge 2–5; ion charge 1–5; 4–15 fragments picked from library; mass accuracy 5.4 ppm; retention time filtering within 10 min predicted RT. Decoys were added using a shuffle method and reintegration was performed with mProphet after search was completed. The final result was reported after filtering peptide detections with a $q$ value cutoff of 0.01.

### Reassignment of proteins reported by each software

Protein groups reported by each software were re-assigned based on razor protein inference proposed by Jürgen et al[19]. Specifically, the shared peptides between mouse and yeast proteins were removed, and the remaining peptides were linked to all matching proteins. For each peptide, its linked proteins were sorted by the number of assigned peptides and the one with the largest number of assigned peptides were selected. These selected proteins were regarded as non-redundant re-assigned proteins reported by specific software. Protein reassignment was only performed for the comparison of protein identification numbers and not for the evaluation of any other metrics.

### Protein intensity determination

Three methods were compared in this study, including Top3 average, $iq$[42] (an R package re-implementing MaxLFQ[43]), and software built-in. Top3 derived protein intensity from intensities of the three most intense precursors. Package $iq$ (version 1.9.1) was implemented in R 4.1.2, with protein groups as the primary ID and precursors as the secondary ID, to estimate protein intensity for each run. Through the comparison of quantification accuracy and consistency for HF and TIMS data (Supplementary Figs. 7, 8), we selected the built-in method for DIA-NN and MaxDIA, and package $iq$ for Spectronaut and Skyline.

### Quantification normalization for synthetic phosphopeptide data analysis

We compared the normalization methods for phosphopeptide quantification. By comparing the quantification linearity of identified phosphopeptides, we chose software-reported quantification for data analysis by DIA-NN and Spectronaut, and AlphaPept[61] implemented normalization function with BFGS for analysis by MaxDIA and Skyline.

### Detection of DEPs and regulated phosphosites

The search result for a benchmark data set was converted to a quantification data matrix which comprised log2 transformed protein intensities from 6 samples and 1 reference, each in five replicates. Any protein with intensity measured in less than three replicates was masked to avoid further processing. Limma[62] (version 3.50.1, with R version 4.1.2) was used to calculate the $p$ value for a protein intensity change in each pairwise comparison, based on linear model fitting and error moderation with an empirical Bayes model (proportion in eBayes at 0.5). Proteins with a fold change (any sample vs reference) >1.5 and BH-adjusted $p$ <0.05 are defined as DEPs. In the TNF-α-induced phosphoproteome data analysis, sequential imputation (ImpSeq) was selected to fill in the NANs as recommended by previous work on DIA phosphoproteome data imputation[63], and phosphosites with intensities in fewer than 3 runs were removed. Then, linear model fitting was performed on the imputed quantification matrix, followed by contrast fitting with the contrast matrix constructed for three experimental groups. Empirical Bayes model was performed with proportion 0.01.

Phosphosites with a fold change >1.5 (TNF-α *vs* DMSO) and BH-adjusted *p* <0.05 (Limma reported) are defined as TNF-α up-regulated sites. Any TNF-α up-regulated sites with a fold change >1.5 (TNF-α *vs* TPCA-1) and BH-adjusted *p* <0.05 (Limma reported) are considered TPCA-1-responding sites.

## Reporting summary

Further information on research design is available in the Nature Portfolio Reporting Summary linked to this article.

## Data availability

Raw MS data generated in this work, spectral libraries, and MS data search reports have been deposited to the ProteomeXchange Consortium[65] via the iProX[66,67] partner repository with the data set identifier PXD034709 (in ProteomeXchange) and IPX0004576000 (in iProX). The human synthetic phosphopeptide data set[14] was downloaded from JPOST with identifier JPST000859; the ProteomeXchange accession code is PXD019797. All raw data, library files, and search results used in this study are summarized in Supplementary Data 1. Source data for main and supplementary figures are provided through Zenodo at https://doi.org/10.5281/zenodo.7409391.

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

## Acknowledgements

We acknowledged professor Jürgen Cox and Brendan MacLean for their helpful discussions on MaxDIA and Skyline software optimization. This work was funded by National Key R&D Program of China (2022YFA1302902 to W.S., 2018YFA0507004 to W.S.), National Natural Science Foundation of China (31971362 to W.S., 32171439 to W.S., 32071245 to Y.Z.), the Natural Science Foundation of Shanghai (20ZR1474400 to Y.Z.), the Shanghai Municipal Science and Technology Major Project (2019SHZDZX02 to Y.Z.), the Shanghai Key Laboratory of Aging Studies (19DZ2260400 to Y.Z.), and Shanghai Frontiers Science Center for Biomacromolecules and Precision Medicine at ShanghaiTech University (to W.S.).

## Author contributions

W.S., Y.Z., and R.L. conceived the project. R.L. performed all data analysis and prepared figures. S.L. prepared proteomics benchmark samples and acquired MS data with the help of X.L. and Y.L. Y.C. prepared samples, acquired MS data, and performed data analysis for the TNF-α-induced phosphoproteomic experiment under the supervision of Y.Z. W.S. and R.L. wrote the manuscript with inputs from S.L., Y.C., and Y.Z.

## Competing interests

The authors declare no competing interests.
