## [Peer Review File · Nature Communications]

REVIEWER COMMENTS

Reviewer #1 (Remarks to the Author):

Lou et al. report a benchmark for data-independent acquisition (DIA) proteomics software. They acquired a two-proteome dataset on two major instrument platforms and evaluated four popular software tools with different spectral library approaches. This was complemented with the analysis of phosphoproteomics data. DIA is certainly on the rise and of great interest to the community. The present study is in the line of the seminal Navarro et al (Nat. Biotech. 2016) paper and complements similar more recent efforts by Froehlich et al (Nat. Commun. 2022), Gotti et al (JPR 2021) and Van Puyvelde et al (Scientific Data 2022). Thus, while technically sound and well written, there is limited novelty in the two-proteome benchmark as even similar timsTOF datasets are now published (albeit not analyzed at this level of detail). That said, I find the phosphoproteomics part of potential interest to a broader audience. Unfortunately, this appears more of an afterthought in the present manuscript and is admittedly much more challenging to benchmark. While this might require additional experiments, I would be supportive of publishing such a revision in Nature Communications.

A few points that the authors might consider in revising their manuscript:

- 1) In evaluating the false discovery rates, the authors observe a very high protein FDR with Skyline. This puts the validity of any comparison using this data into question and the authors might consider removing Skyline protein data from the main figures.
- 2) Fig. 4/5A show indicate re-assigned protein numbers for comparison.
- 3) Please indicate in the main text which statistical model was used to find differentially expressed proteins.
- 4) The authors should indicate whether any attempts at contacting the software developers were taken to find optimized settings (similar to Navarro et al.).
- 5) The phospho benchmark is very interesting, but also limited by the small N of phosphopeptides. Hogrebe et al (<https://pubmed.ncbi.nlm.nih.gov/29535314/>) have recently published a strategy to

benchmark quantification strategies for phosphoproteomics. Could the authors conceive/perform similar experiments for their purposes?

6) Given that the authors highlight the two instrument platforms in the other sections, do they expect any differences for timsTOF phospho data?

7) The authors stress the importance of library generation, however, in the phosphoproteomics benchmark only a project-specific DDA library is used. How would the results change for the DDA-independent library case?

Reviewer #2 (Remarks to the Author):

In this work, the authors performed a single-center benchmarking study investigating the performance of four mainstream DIA software suites (DIA-NN, MaxDIA, Skyline, and Spectronaut) and suggested two software DIA-NN and MaxDIA perform better than Skyline and MaxDIA. Whereas many recent DIA papers do not use Skyline for large-scale identification and quantification, according to the author's report, it is indeed surprising that MaxDIA performed significantly worse than the other two, even though MaxDIA was published recently in a Nature Biotech paper and demonstrated a comparable or even better performance than Spectronaut by that time.

Although we appreciate the results shared by the authors and agree that this work should be interesting to some DIA users and should be published in some journals, we doubt its significance for Nature Communications. The limited biological/ clinical scope of their work also seems to be more relevant to a specialized journal. Please see below for more discussions.

Inconsistency issue. In the original MaxDIA publication, comparable or even better results than Spectronaut were shown for both peptide identification and quantification. It is unclear why the authors discovered inconsistent results (especially for proteome level analysis). Of course, data is data. We therefore suggest that, instead of only analyzing their own data, the authors should consider analyzing the same datasets in previous publications, including the MaxDIA paper, such as the three-species dataset from LFQbench (Navarro et al 2016) and the four-species dataset from Bruderer et al 2017, both were widely used in many following benchmark studies, to compellingly illustrate the inconsistent performance reported.

User experience vs. developer view. Although reporting the difference between software suites from a user's perspective is well justified, for the community to grow, it is much more useful to understand why different results occurred and how each software can be optimized. In this regard, Navarro et al serves a nice example in which all the developers were involved in that study to reanalyze the inconsistent data and optimize their respective tools towards a highly convergent result. In some cases, the refining steps can be minor and dataset-dependent. In this way, the software developers (such as MaxDIA in this case) would feel fairer as well, with their work properly evaluated by the entire community.

As another critical consideration, because the four software suites are heavily developed respectively (note this can be different in other areas), their performance may not be stabilized for a user to compare. Just as an example, Biognosys released Spectronaut 16 recently, which is reported to achieve even better results in proteome identification than DIA-NN for all gradient lengths, whereas the work presented here used Spectronaut 15 in their comparison, making their results less interesting to the current Spectronaut users. The users are therefore less likely to treat the present result as a "practice guide". It is almost certain that DIA-NN will also get optimized in another new version soon. At least the author should mention the versions of the software tools in one of their main figures.

Phosphoproteomics DIA. The phosphoproteomic work presents a highlight of this study. Although the authors analyzed a biological dataset in Reference 45, they analyzed the results minimally. Only identification numbers and quantitative CVs were compared. The largest challenge in phosphoproteomics DIA lies in the confident localization of phosphorylation sites between samples in a large cohort and how that would affect the biological studies (such as the kinase motif interference, which is based on phosphosite localization) if different software tools are used. In this regard, the scope of application in the present study is not as comprehensive as the recent study by Frohlich et al. *Nature Communications*, 2022. Therefore, the authors are encouraged to expand the evaluation of data analysis for phosphoproteomics DIA.

Dear reviewers,

We deeply appreciated your efforts in reviewing our manuscript and providing very insightful comments. We have substantially revised our manuscript according to your suggestion. The major revisions have been made in the following aspects:

1. Because Spectronaut evaluated in our study recently released a new version (v16) with significant improvement, we re-analyzed our benchmark data sets with four software suites all in their latest versions. In addition, we compared three versions of Spectronaut (v14, v15 and v16) in the benchmark data analysis because published studies mainly used the previous versions such as v13 and v14. It took us quite a while to finish all data searches, and we updated results in the text, main figures and supplementary information, which led to changes of certain conclusions.
2. We substantially expanded the DIA phosphoproteomics data analysis as suggested by the reviewers. Specifically, we added the site-specific phosphorylation stoichiometry analysis to benchmark quantification performance. We also added phosphopeptide data analysis by different workflows using a DDA-independent library to be compared with previous results from a project-specific DDA library. Furthermore, to illustrate how different DIA workflows affect phosphoproteomics data analysis in a biological setting, we performed a new experiment in which cells were treated with TNF- α in the absence or presence of an I-kappa-B kinase inhibitor, and we acquired DIA data on both QE HF-X and timsTOF Pro instruments. With these new data sets, we systematically compared the phosphoproteome coverage, number of regulated phosphosites, and enrichment of signaling pathways and kinases based on DIA data analysis with different software and different libraries. The results of the TNF- α -induced phosphoproteome analysis were presented in a new session (containing a new figure 5) while the original sessions for global proteomics were shortened so that the two main subjects of this study (DIA proteomics and phosphoproteomics) are more balanced.
3. We carried out extensive parameter optimization and tests for each software using several benchmark data sets to highlight critical settings, which are summarized in Supplementary Notes 1-4 added in revision.

The major changes are marked in red in our revised manuscript and a point-by-point response letter is provide below.

We thank you very much again for reviewing our manuscript.

Yours sincerely,

Wenqing Shui,
Research Professor
iHuman Institute
ShanghaiTech University
Shanghai, China

REVIEWER COMMENTS

Reviewer #1 (Remarks to the Author):

Lou et al. report a benchmark for data-independent acquisition (DIA) proteomics software. They acquired a two-proteome dataset on two major instrument platforms and evaluated four popular software tools with different spectral library approaches. This was complemented with the analysis of phosphoproteomics data. DIA is certainly on the rise and of great interest to the community. The present study is in the line of the seminal Navarro et al (Nat. Biotech. 2016) paper and complements similar more recent efforts by Froehlich et al (Nat. Commun. 2022), Gotti et al (JPR 2021) and Van Puyvelde et al (Scientific Data 2022). Thus, while technically sound and well written, there is limited novelty in the two-proteome benchmark as even similar timsTOF datasets are now published (albeit not analyzed at this level of detail). That said, I find the phosphoproteomics part of potential interest to a broader audience. Unfortunately, this appears more of an afterthought in the present manuscript and is admittedly much more challenging to benchmark. While this might require additional experiments, I would be supportive of publishing such a revision in Nature Communications.

RESPONSE: We thank the reviewer very much for his/her excellent and encouraging comments, and agreed with the reviewer that our study design of DIA proteomics data analysis overlapped in part with the previous benchmark studies. But it is also noteworthy that during revision, we re-analyzed our benchmark data using the latest versions of all software suites to be evaluated, which generated results different from those obtained earlier, especially for Spectronaut. While the recently published studies by Froehlich et al (Nat. Commun. 2022) and Gotti et al (JPR 2021) used Spectronaut v14, the latest version (v16) used in our study significantly increased the proteome coverage yet showed a trade-off in quantification performance. In addition, we scanned through key parameters in each software to find out optimal settings with our benchmark data, which are elaborated in newly added Supplementary Notes 1-4. Hence, our benchmark study would provide a practical guidance as to how to construct DIA data analysis workflows for global proteomics.

Nevertheless, as kindly pointed out the reviewer, we do realize the phosphoproteomics part need to be strengthened. Thus, we performed a new TNF- α -induced phosphoproteomics experiment and substantially expanded the DIA phosphoproteomics data analysis. Taken together, we hope our extensive revision could meet the reviewer's expectation.

A few points that the authors might consider in revising their manuscript:

- 1) In evaluating the false discovery rates, the authors observe a very high protein FDR with Skyline. This puts the validity of any comparison using this data into question and the authors might consider removing Skyline protein data from the

main figures.

RESPONSE: We thank the reviewer for bringing up this critical point. Because Skyline lacks FDR control at the protein level, it typically reports more false protein identifications than other software suites when assessed with a decoy library. But Skyline does have its unique strengths such as high performance in proteomic quantification (Fig. 3). So we would like to keep the protein-level results by Skyline in the figures, and added an indication of the FDR problem (Fig. 2a). The relevant FDR data was moved to Supplementary Fig. 6b.

2) Fig. 4/5A show indicate re-assigned protein numbers for comparison.

RESPONSE: When using the latest versions of four software, we found the number of protein identifications re-assigned based on razor protein inference was very close to that reported in search results by different software (Supplementary Fig. 2). So we used the software reported protein numbers for comparison, and added a notion to the revised manuscript:

“Although each software may assemble protein groups in a different way, we found the number of protein identifications re-assigned based on razor protein inference¹⁹ was very close to that reported in search results which were then used directly for comparison (Supplementary Fig. 2).” (Page 7-8)

3) Please indicate in the main text which statistical model was used to find differentially expressed proteins.

RESPONSE: Limma was used and it is indicated in the main text of the revised manuscript (Pages 14, 20). In fact, we compared several commonly used models (Student's t-test, Welch's t-test, SAM and Limma) and showed the results in Supplementary Note 5.

4) The authors should indicate whether any attempts at contacting the software developers were taken to find optimized settings (similar to Navarro et al.).

RESPONSE: Because in our original work, MaxDIA and Skyline showed generally lower performance than the other two software tools, we contacted their developers and discussed how to optimize software parameters.

For MaxDIA optimization, we first tried tuning several parameters and found the transfer q value was the most critical one. Then we scanned through five transfer q values, yet did not observe large improvement of quantification performance. After contacting Prof. Jürgen Cox, we received an advice to try a new version they just released (v2.1.3). The new version did improve the quantification accuracy and precision in analyzing one of our benchmark data sets (Supplementary Note 4.3). So we updated all results with v2.1.3 in revision. We also sent the major result to Prof. Jürgen Cox, and he confirmed that our settings were suitable and his team would continue improving MaxDIA.

For Skyline, we contacted Brendan MacLean in the developer team. According to his

suggestion, we evaluated the influence of different MS2 mass error tolerances in the analysis of several benchmark data sets to find optimal settings (Supplementary Note 3). The developer also confirmed our proper settings in Skyline.

5) The phospho benchmark is very interesting, but also limited by the small N of phosphopeptides. Hoglebe et al (<https://pubmed.ncbi.nlm.nih.gov/29535314/>) have recently published a strategy to benchmark quantification strategies for phosphoproteomics. Could the authors conceive/perform similar experiments for their purposes?

RESPONSE: We are very thankful to the reviewer's excellent suggestion. To benchmark the quantification performance of different DIA workflows, we analyzed the data set from a hybrid phosphoproteome sample prepared with fixed phosphopeptide stoichiometries ranging from 1% to 99% using the strategy published by Hoglebe et al. After extracting the quantification data for phosphopeptides, non-phosphopeptides and corresponding protein intensities, we implemented a 3D multiple regression model-based approach for site-specific stoichiometry calculation as previously described¹. Although phosphopeptide identification from this data set by Spectronaut outnumbered that by DIA-NN as expected, stoichiometry calculation based on quantification results by both software using a project-specific DDA library yielded equally high accuracy and similar precision for all stoichiometry levels and across a wide range of phosphosite score cut-offs (Supplementary Fig. 14).

1. Bekker-Jensen, D.B. et al. Rapid and site-specific deep phosphoproteome profiling by data-independent acquisition without the need for spectral libraries. *Nat Commun* 11, 787 (2020).

Supplementary Figure 14 Phosphorylation stoichiometry measurement using a benchmark data set.

(a) Number of identified yeast and human total peptides and phosphopeptides by DIA-NN or Spectronaut with a project-specific DDA library. **(b)** Number of identified phosphopeptides as a function of the phosphosite confidence score cut-off by DIA-NN (blue line) or Spectronaut (red line). **(c)** Boxplot of calculated site phosphorylation stoichiometry based on DIA data analysis by DIA-NN (left) or Spectronaut (right). **(d)** Distribution of calculated phosphorylation stoichiometry as a function of the phosphosite confidence score cut-off by DIA-NN (blue shade) or Spectronaut (red shade) under each condition. Median values are indicated by blue and red lines.

We added this phosphorylation stoichiometry analysis to the revised manuscript (Pages 18-19).

6) Given that the authors highlight the two instrument platforms in the other sections, do they expect any differences for timsTOF phospho data?

RESPONSE: Again we thank the reviewer for his/her critical comment. To address this issue, we performed a new experiment in which cells were treated with TNF- α in the absence or presence of an I-kappa-B kinase inhibitor, and we acquired DIA data on both QE HF-X and timsTOF Pro instruments. With these two data sets, we systematically compared the phosphoproteome coverage, number of regulated phosphosites, and enrichment of signaling pathways based on DIA data analysis with different software and different libraries. The results and main conclusions can be found in the new session of "DIA phosphoproteomics data analysis in a biological setting" (Pages 19-21). In brief, "our analysis based on TNF- α -regulated sites indicated the preference of DIA-NN in processing HF-X phosphoproteomics data and Spectronaut in processing TIMS data. Furthermore, the use of a DDA-independent library by both software exhibited comparable or even slightly better performance than the project-specific DDA library in the identification of known TNF- α -dependent phosphosites and signaling pathways, probably due to the relatively small size of the DDA library generated in this study."

7) The authors stress the importance of library generation, however, in the phosphoproteomics benchmark only a project-specific DDA library is used. How would the results change for the DDA-independent library case?

RESPONSE: To address this key question kindly raised by the reviewer, we analyzed the synthetic phosphopeptide data and the TNF- α -induced phosphoproteome data with both a project-specific DDA library and a DDA-independent library. For the synthetic phosphopeptide data, Spectronaut and DIA-NN using two types of libraries showed very similar trends in the sensitivity and global FDR tests (Fig. 4 and Supplementary Fig. 13). For the TNF- α -induced phosphoproteome data, the use of a DDA-independent library by both software exhibited comparable or even slightly better performance than the project-specific DDA library in the identification of known TNF- α -dependent phosphosites and signaling pathways (Fig. 5). More detailed results can be found in the revised manuscript (Pages 18-21).

Reviewer #2 (Remarks to the Author):

In this work, the authors performed a single-center benchmarking study investigating the performance of four mainstream DIA software suites (DIA-NN, MaxDIA, Skyline, and Spectronaut) and suggested two software DIA-NN and MaxDIA perform better than Skyline and MaxDIA. Whereas many recent DIA papers do not use Skyline for large-scale identification and quantification, according to the author's report, it is indeed surprising that MaxDIA performed significantly worse than the other two, even though MaxDIA was published recently in a Nature Biotech paper and demonstrated a comparable or even better performance than Spectronaut by that time.

Although we appreciate the results shared by the authors and agree that this work should be interesting to some DIA users and should be published in some journals, we doubt its significance for Nature Communications. The limited biological/ clinical scope of their work also seems to be more relevant to a specialized journal. Please see below for more discussions.

Inconsistency issue. In the original MaxDIA publication, comparable or even better results than Spectronaut were shown for both peptide identification and quantification. It is unclear why the authors discovered inconsistent results (especially for proteome level analysis). Of course, data is data. We therefore suggest that, instead of only analyzing their own data, the authors should consider analyzing the same datasets in previous publications, including the MaxDIA paper, such as the three-species dataset from LFQbench (Navarro et al 2016) and the four-species dataset from Bruderer et al 2017, both were widely used in many following benchmark studies, to compellingly illustrate the inconsistent performance reported.

RESPONSE: We thank the reviewer most deeply for making these excellent comments which definitely helped us improve our work. Indeed we were also surprised to see the lower performance of MaxDIA than the other software tools to be evaluated. As kindly suggested by the reviewer, we re-analyzed the LFQbench data using the same version of MaxDIA (v2.0.0) published in *Nat Biotechnol* 39, 1563-1573 (2021). As shown in Fig. R1, we were able to obtain results blue consistent with the original publication.

Fig. R1. Comparison of protein identification and quantification results of the LFQBench data by MaxDIA (v2.0.0) from the original publication (left) and our own re-analysis (right).

In addition, we tried tuning several parameters in MaxDIA and found the transfer q value was the most critical one. Then we scanned through five transfer q values, yet did not observe significant improvement of its quantification performance (Supplementary Note 4.1).

It is noteworthy that in the MaxDIA paper, the authors used two earlier versions of Spectronaut for comparison (v13 and v14). However, Spectronaut has undergone significant changes in the most recent releases (v15 and v16, see Supplementary Note 1 for version comparison). During revision, we re-analyzed all benchmark data with four software suites all in their latest versions. Based on these updated results, we revised our conclusion as we noticed that MaxDIA (v2.1.3) and Spectronaut (v16) both yielded larger quantification variations than DIA-NN and Skyline, though Spectronaut v16 increased the proteome coverage compared to v15 (Fig. 3, Supplementary Note 1). We sent the major result to Prof. Jürgen Cox (MaxDIA developer team leader), and he confirmed that our software settings were suitable and his team would continue improving MaxDIA.

As for Skyline, it is a classical MS data processing tool with great performance in peptide peak extraction and quantification. Its open-access feature, transparent framework and ability to process raw data from almost all types of MS instruments makes it still very impactful in the proteomic community. We also contacted the Skyline developer and received suggestions to optimize key parameters (Supplementary Note 3).

Taken together, we have substantially expanded our study of DIA proteomics and phosphoproteomics data analysis in revision, and hope this work would provide a useful guidance to both software users and developers.

User experience vs. developer view. Although reporting the difference between software suites from a user's perspective is well justified, for the community to grow, it is much more useful to understand why different results occurred and how each software can be optimized. In this regard, Navarro et al serves a nice example in which all the developers were involved in that study to reanalyze the inconsistent data and optimize their respective tools towards a highly convergent result. In some cases, the refining steps can be minor and dataset-dependent. In this way, the software developers (such as MaxDIA in this case) would feel fairer as well, with their work properly evaluated by the entire community.

RESPONSE: Because in our original work, MaxDIA and Skyline showed generally lower performance than the other two software tools, we contacted their developers

and discussed how to optimize software parameters.

For MaxDIA optimization, after contacting Prof. Jürgen Cox, we received an advice to try a new version they just released (v2.1.3). By comparing the results from v2.0.3 and v2.1.3, we found the new version improved the quantification accuracy and precision in analyzing one of our benchmark data sets (Supplementary Note 4.3).

For Skyline, as suggested by the developer, we specifically optimized the MS2 mass tolerance for HF and TIMS data analysis, which led to increased proteome coverage while maintaining the quantification performance (Supplementary Note 3). Finally, we reached consensus with both MaxDIA and Skyline developers that we properly used their software in this study to generate convincing results. Our extensive efforts in each software optimization are summarized in Supplementary Notes 1-4. We also briefly discussed the possible reasons underlying different software performance (Pages 23-24, Supplementary Notes).

As another critical consideration, because the four software suites are heavily developed respectively (note this can be different in other areas), their performance may not be stabilized for a user to compare. Just as an example, Biognosys released Spectronaut 16 recently, which is reported to achieve even better results in proteome identification than DIA-NN for all gradient lengths, whereas the work presented here used Spectronaut 15 in their comparison, making their results less interesting to the current Spectronaut users. The users are therefore less likely to treat the present result as a “practice guide”. It is almost certain that DIA-NN will also get optimized in another new version soon. At least the author should mention the versions of the software tools in one of their main figures.

RESPONSE: We thank the reviewer for raising this very critical point. We fully agree with the reviewer’s notion and added the following statement to Discussion (Page 25): “we recognized the four software suites are undergoing intensive and continuous development, and their respective stronger or weaker performances may change in the future.”

Indeed the newly released Spectronaut v16 showed significant improvement in proteome identification compared to v15. The developer also reported a higher proteome coverage using v16 than DIA-NN on Spectronaut website. So we re-analyzed several benchmark data sets generated on our own or from other labs with four software suites all in their latest versions (annotated in Figure 1b), and updated results in the text and supplementary information. Interestingly, with our data sets, we did not see an obvious advantage of Spectronaut (v16) over DIA-NN (v1.8.1) in the proteome coverage. While Spectronaut reported slightly more mouse protein identifications with the universal or DDA-dependent libraries, DIA-NN achieved a higher proteome coverage with a DDA-independent library (Fig. 2). Moreover, DIA-NN showed better performance than Spectronaut in regard to the false positive/negative rate control, quantification accuracy and precision as well as sensitivity and specificity of DEP detection for most comparisons (Figs. 2, 3). Given the overall superior

performance and the open-access feature, we would recommend DIA-NN for global DIA proteomics data analysis (Page 23).

Furthermore, evaluation of DIA-NN (v1.8.1) and Spectronaut (v16) in processing DIA phosphoproteomics data clearly revealed the pros and cons of each software not reported before (Pages 16-21). We highlighted several key aspects for users to consider when constructing their own DIA phosphoproteomic data analysis workflows in Discussion (Pages 24-25). Therefore, by conducting a comprehensive and unbiased comparison of mainstream DIA data analysis workflows using multiple benchmark data sets, our study is expected to help users design a robust workflow as well as the developers improve the software from a user point of view.

Phosphoproteomics DIA. The phosphoproteomic work presents a highlight of this study. Although the authors analyzed a biological dataset in Reference 45, they analyzed the results minimally. Only identification numbers and quantitative CVs were compared. The largest challenge in phosphoproteomics DIA lies in the confident localization of phosphorylation sites between samples in a large cohort and how that would affect the biological studies (such as the kinase motif interference, which is based on phosphosite localization) if different software tools are used. In this regard, the scope of application in the present study is not as comprehensive as the recent study by Frohlich et al. Nature Communications, 2022. Therefore, the authors are encouraged to expand the evaluation of data analysis for phosphoproteomics DIA.

RESPONSE: Again we are very grateful to the reviewer for making this excellent comment. To illustrate how different DIA workflows affect phosphoproteomics data analysis in a real biological setting, we performed a new experiment in which cells were treated with TNF- α in the absence or presence of an I-kappa-B kinase inhibitor, and we acquired DIA data on both QE HF-X and timsTOF Pro instruments. With these new data sets, we systematically compared the phosphoproteome coverage, number of regulated phosphosites, and enrichment of signaling pathways based on DIA data analysis with different software and different libraries (see the new session added to Results, Pages 19-21). Comparison of DIA-NN and Spectronaut in the TNF- α -induced phosphoproteomics analysis revealed their differential behaviors in the enrichment of signaling pathways and discovery of known regulated phosphosites from HF-X data *versus* TIMS data (Fig. 5). Together with other additional analysis of site phosphorylation stoichiometry and with DDA-independent libraries (Pages 18-19), we substantially expanded the DIA phosphoproteomics data analysis as kindly suggested by both reviewers.

REVIEWERS' COMMENTS

Reviewer #1 (Remarks to the Author):

I appreciate the authors' effort to revise their manuscript substantially. Important changes in response to the reviewers are 1) re-analysis of the entire data set with newer software versions and optimized parameters and 2) extension of the phosphoproteomics part with an additional experiment. A few minor remarks remain:

1) The tone in the discussion section appears somewhat defensive as the authors aim to set their study apart from prior work. I would recommend to give this section a more appreciative and positive spin.

2) Please clarify in the figure legends whether identifications refer to yeast, mouse or both proteomes.

3) 'Given that mouse proteins accounted for only 5%-40% of total proteins in the hybrid proteome samples with'. – This should read '5-40% of total protein mass'.

4) There are a few more typos and grammar errors that should be addressed by careful editing in the next steps.

5) Title: 'data mining' does not describe the study well and could be omitted from the title.

6) Wording: replace 'mainstream' by 'popular'?

Reviewer #2 (Remarks to the Author):

Despite the present revision in which the authors included an updated analysis of a new version of Spectronaut (v16) and a new simple phosphoproteomics-DIA experiment, our previous main

critiques remain. These conservations include (1) The difficulty of comparing software tools of different versions– for example, Spectronaut is having v17 released for tests. (2) The scope of a user-based comparison between DIA tools (instead of a community-wide effort) is rather limited, which better suits a specialized journal. (3) The new phosphoproteomics data still lack technical or biological golden standards, and their data evaluation did not incorporate e.g., the evaluation of phosphosite localization in complex samples and how missing value problem would impact phosphoproteomic data analysis (both localization and quantification) across large-scale studies. The above being said, the authors indeed raised considerable efforts in comparing the software suites, which would provide helpful feedback to software developers.

REVIEWERS' COMMENTS

Reviewer #1 (Remarks to the Author):

I appreciate the authors' effort to revise their manuscript substantially. Important changes in response to the reviewers are 1) re-analysis of the entire data set with newer software versions and optimized parameters and 2) extension of the phosphoproteomics part with an additional experiment. A few minor remarks remain:

1) The tone in the discussion section appears somewhat defensive as the authors aim to set their study apart from prior work. I would recommend to give this section a more appreciative and positive spin.

RESPONSE: We thank the reviewer very much for pointing out this problem of expression, and have rephrased this part in Discussion in a milder and more positive tone to show our appreciation of the previous work. On the other hand, we still feel it helpful to list a few features of our study that differ from previous ones such that the readers could have an overview of the major findings from our study.

2) Please clarify in the figure legends whether identifications refer to yeast, mouse or both proteomes.

RESPONSE: We thank the reviewer for his/her great comment, and have revised the figure legends accordingly.

3) 'Given that mouse proteins accounted for only 5%-40% of total proteins in the hybrid proteome samples with' . - This should read '5-40% of total protein mass' .

RESPONSE: It has been changed to '5-40% of total protein mass' in the revised manuscript.

4) There are a few more typos and grammar errors that should be addressed by careful editing in the next steps.

RESPONSE: We are very sorry for these typos and errors, and have corrected them as much as possible in revision.

5) Title: 'data mining' does not describe the study well and could be omitted from the title.

RESPONSE: We agree with the reviewer and deleted it from the title.

6) Wording: replace 'mainstream' by 'popular' ?

RESPONSE: We replaced "mainstream" by 'commonly used' in the title and the text.

Reviewer #2 (Remarks to the Author):

Despite the present revision in which the authors included an updated analysis of a new version of Spectronaut (v16) and a new simple phosphoproteomics-DIA experiment, our previous main critiques remain. These conservations include (1) The difficulty of comparing software tools of different versions - for example, Spectronaut is having v17 released for tests. (2) The scope of a user-based comparison between DIA tools (instead of a community-wide effort) is rather limited, which better suits a specialized journal. (3) The new phosphoproteomics data still lack technical or biological golden standards, and their data evaluation did not incorporate e.g., the evaluation of phosphosite localization in complex samples and how missing value problem would impact phosphoproteomic data analysis (both localization and quantification) across large-scale studies. The above being said, the authors indeed raised considerable efforts in comparing the software suites, which would provide helpful feedback to software developers.

RESPONSE: We thank the reviewer very much for his/her excellent comments. As a panel of DIA software tools are available and they are continuously being upgraded, we feel that an objective and comprehensive evaluation from the user perspective would be of high values to the community. All our benchmark data, spectral libraries and MS data search reports are downloadable from the public repository, which guarantees the transparency and reproducibility of our results. With respect to the phosphoproteomics data analysis, our study in fact evaluated the influence of varying the phosphosite localization confidence on the identification depth, enrichment of known TNF- α signaling pathways and known TNF- α -regulated phosphosites by different software (Figure 5). Because there is no absolute “ground-truth” for any real-case phosphoproteomics data from complex samples, we evaluate phosphosite localization and phosphosite/phosphoprotein quantification by comparison of known regulated phosphosites and pathways in the biological context. So far this is the most accepted way of “true phosphosite” validation for complex proteomics data.

We highly appreciated the reviewer’s comment on the missing value problem and prepared two new Supplementary Figures to address this point. First, we assessed the completeness of detected phosphosites as a function of the site confidence from the original data without missing value imputation (Supplementary Figure 16, also see below). Second, we compared the bioinformatics analysis results based on the original data (Supplementary Figure 17, see below) *vs* the imputed data (Figure 5) to find out more known signaling pathways and more known TNF- α -responding phosphosites in line with a higher number of total regulated phosphosites can be recovered from the imputed data, suggesting missing value imputation using an appropriate algorithm could facilitate DIA phosphoproteomics analysis. These new results are added to the revised manuscript (Page 21).

Supplementary Figure 16 Completeness of detected phosphosites in TNF- α -induced phosphoproteomic analysis without phosphosite intensity imputation. The number of phosphosites identified from N replicates at a specific condition (N=1,2,3) is plotted as a function of the site confidence.

a, Result from HF-X data. b, Result from TIMS data.

Supplementary Figure 17 Comparison of DIA-NN and Spectronaut in TNF- α -induced phosphoproteome data analysis without phosphosite intensity imputation

a, Number of TNF- α -regulated phosphosites from the analysis of HF-X data (left) and TIMS data (right) with different workflows. Phosphosites in response to TPCA-1 are those up-regulated by TNF- α and suppressed by TPCA-1 treatment, and their percentages over all up-regulated sites by TNF- α are indicated. b, Enriched KEGG pathways based on the analysis of HF-X data (left) and TIMS data (right) with different workflows. Significantly enriched pathways (adjusted $p < 0.05$) are annotated in a color gradient. c, Phosphosites up-regulated by TNF- α and included in the TNF- α pathway. They were identified in the

analysis of HF-X data (left) and TIMS data (right) with different workflows. Significantly regulated sites (FC >1.5, adjusted p <0.05) are annotated in a color gradient.